Hammerhead flatworms (Platyhelminthes, Geoplanidae, Bipaliinae): mitochondrial genomes and description of two new species from France, Italy, and Mayotte

http://orcid.org/0000-0002-7155-4540 Justine Jean-Lou 1 justine@mnhn.fr
http://orcid.org/0000-0001-8661-5118 Gastineau Romain 2
Gros Pierre 3
Gey Delphine 4
http://orcid.org/0000-0003-1020-1247 Ruzzier Enrico 5
Charles Laurent 6
Winsor Leigh 7
1 ISYEB-Institut de Systématique, Évolution, Biodiversité, Muséum National d’Histoire Naturelle , Paris , France
2 Institute of Marine and Environmental Sciences, University of Szczecin , Szczecin , Poland
3 Amateur Naturalist, Unaffiliated , Cagnes-sur-Mer , France
4 Molécules de Communication et Adaptation des Micro-Organismes, Muséum National d’Histoire Naturelle , Paris , France
5 Department of Agronomy, Food, Natural Resources, Animals and the Environment (DAFNAE) , Padova , Italy
6 Muséum de Bordeaux - science et nature , Bordeaux , France
7 James Cook University , Townsville, Queensland , Australia
Riutort Marta
Electronic publication date: 2022 Feb 1
Publication date: 2022
Volume: 10
Electronic Location ID: e12725
Received 2021 Aug 18; Accepted 2021 Dec 10
Copyright: © 2022 Justine et al.
Copyright year: 2022
Copyright holder: Justine et al.
License: This is an open access article distributed under the terms of the Creative Commons Attribution License, which permits unrestricted use, distribution, reproduction and adaptation in any medium and for any purpose provided that it is properly attributed. For attribution, the original author(s), title, publication source (PeerJ) and either DOI or URL of the article must be cited.
License URL: https://creativecommons.org/licenses/by/4.0/

Keywords: Platyhelminthes, Land planarians, Alien invasive species, France, Mayotte, Italy, Barcoding, Citizen science, Mitogenome, Taxonomy

Funding: Muséum National d’Histoire Naturelle, Paris, France This work was supported by several grants (ATM) from the Muséum National d’Histoire Naturelle, Paris, France. The funders had no role in study design, data collection and analysis, decision to publish, or preparation of the manuscript.

==============================
Background

New records of alien land planarians are regularly reported worldwide, and some correspond to undescribed species of unknown geographic origin. The description of new species of land planarians (Geoplanidae) should classically be based on both external morphology and histology of anatomical structures, especially the copulatory organs, ideally with the addition of molecular data.

Methods

Here, we describe the morphology and reproductive anatomy of a species previously reported as Diversibipalium “black”, and the morphology of a species previously reported as Diversibipalium “blue”. Based on next generation sequencing, we obtained the complete mitogenome of five species of Bipaliinae, including these two species.

Results

The new species Humbertium covidum n. sp. (syn: Diversibipalium “black” of Justine et al., 2018) is formally described on the basis of morphology, histology and mitogenome, and is assigned to Humbertium on the basis of its reproductive anatomy. The type-locality is Casier, Italy, and other localities are in the Department of Pyrénées-Atlantiques, France; some published or unpublished records suggest that this species might also be present in Russia, China, and Japan. The mitogenomic polymorphism of two geographically distinct specimens (Italy vs France) is described; the cox1 gene displayed 2.25% difference. The new species Diversibipalium mayottensis n. sp. (syn: Diversibipalium “blue” of Justine et al., 2018) is formally described on the basis of external morphology and complete mitogenome and is assigned to Diversibipalium on the basis of an absence of information on its reproductive anatomy. The type- and only known locality is the island of Mayotte in the Mozambique Channel off Africa. Phylogenies of bipaliine geoplanids were constructed on the basis of SSU, LSU, mitochondrial proteins and concatenated sequences of cox1, SSU and LSU. In all four phylogenies, D. mayottensis was the sister-group to all the other bipaliines. With the exception of D. multilineatum which could not be circularised, the complete mitogenomes of B. kewense, B. vagum, B. adventitium, H. covidum and D. mayottensis were colinear. The 16S gene in all bipaliine species was problematic because usual tools were unable to locate its exact position.

Conclusion

Next generation sequencing, which can provide complete mitochondrial genomes as well as traditionally used genes such as SSU, LSU and cox1, is a powerful tool for delineating and describing species of Bipaliinae when the reproductive structure cannot be studied, which is sometimes the case of asexually reproducing invasive species. The unexpected position of the new species D. mayottensis as sister-group to all other Bipaliinae in all phylogenetic analyses suggests that the species could belong to a new genus, yet to be described.

Introduction

Many new records of alien land planarians (Geoplanidae) have been published in recent years; some correspond to already known species found in new locations, but some are in fact undescribed species, never mentioned in other countries and for which the location of origin is unknown. Recent typical examples include Obama nungara Carbayo et al., 2016, a species from South America now invasive in Europe, for which taxonomic confusion has obscured the debate over the last decade (Carbayo et al., 2016; Justine, Thévenot & Winsor, 2014; Lago-Barcia et al., 2015) and Caenoplana decolorata Mateos et al., 2020, probably from Australia (Justine et al., 2020b; Mateos et al., 2020). In addition to the scientific need for precision, it is important to ascribe precise binomial names to invasive species for administrative purposes.

Although some land planarians show brilliant colours and patterns, these are generally not sufficient for describing species. A formal description should classically be based on both external morphology and microanatomy including that of the copulatory organs, obtained by histological techniques. Modern descriptions generally add partial sequences of a few genes. However, some invasive species have abandoned sexual reproduction and thus lack most characters usable for taxonomy.

Hammerhead flatworms (subfamily Bipaliinae) are among the most spectacular land flatworms, with one species, Bipalium nobile Kawakatsu & Makino, 1982, reaching one metre in length (see Table 1 for authors of taxa and key references). In a taxonomic revision of the Bipaliinae, a collective group, the genus Diversibipalium Kawakatsu et al., 2002, was erected to accommodate uncertain bipaliid species that had descriptions based on immature specimens, or mature specimens whose internal anatomy, including that of the copulatory organs, has not yet been investigated (Kawakatsu et al., 2002). All the species assigned to this collective genus are described only on the basis of external morphology and colour pattern. In 2018, we reported two species and assigned them to Diversibipalium, but we did not create binomial names; instead, the species were designated as Diversibipalium “black”, found in a single place in France, and Diversibipalium “blue”, found only on the island of Mayotte, off Africa (Justine et al., 2018).

Table 1 Hammerhead flatworms (Geoplanidae, Bipaliinae), authors of taxa and key references for biology and mitogenome.

Taxon and authors	Reference for taxon	Main references for biology	Mitochondrial genome	
Bipalium kewense Moseley, 1878	Moseley, 1878	Winsor, 1983	Gastineau et al., 2019	
Bipalium vagum Jones & Sterrer, 2005	Jones & Sterrer, 2005	Ducey, McCormick & Davidson, 2007	This paper	
Bipalium adventitium Hyman, 1943	Hyman, 1943	Ducey et al., 2005	This paper	
Bipalium pennsylvanicum Ogren, 1987	Ogren, 1987	Ogren & Sheldon, 1991	Unknown	
Diversibipalium multilineatum (Makino & Shirasawa, 1983) Kubota & Kawakatsu, 2010	Makino & Shirasawa, 1983	Makino & Shirasawa, 1986	This paper	
Humbertium covidum n. sp.	This paper	Justine et al., 2018; This paper	This paper	
Diversibipalium mayottensis n. sp.	This paper	Justine et al., 2018; This paper	This paper	
Note:

The list includes the main invasive taxa and the species studied here.

Histology is a technique that requires specialised skills and experience in interpreting sections. In contrast, the development of next generation sequencing technologies (NGS) has made it easier and cheaper to obtain sequences. It is now possible to sequence the organellar genomes of various organisms with a satisfactory rate of success. For land planarians, the first complete mitogenome was described in 2015 (Solà et al., 2015) from a specimen of Obama sp., since then considered to be Obama nungara Carbayo et al., 2016 (Carbayo et al., 2016). Recently, our group supplemented the databases with complete mitogenomes from several other invasive species, namely Bipalium kewense (Gastineau et al., 2019), Platydemus manokwari de Beauchamp, 1963 (Gastineau et al., 2020), Parakontikia ventrolineata (Dendy, 1892) (Gastineau & Justine, 2020), and Amaga expatria Jones & Sterrer, 2005 (Justine et al., 2020a).

Complete mitogenomes provide a different type of data for molecular identification and phylogeny when compared with the usual molecular markers such as the short and long subunits of the nuclear ribosomal RNA genes (SSU and LSU, respectively). Indeed, the organisation of the mitogenome itself, like gene order, gene composition and the presence of pseudo-genes, can provide an additional phylogenetic signal. Based on our previous experience, phylogenetic trees inferred from mitochondrial protein-coding gene alignments display strong support at their nodes, congruent with taxonomy, classification and biogeography (Justine et al., 2020a). Also, the use of next generation sequencing rather than PCR coupled with Sanger to obtain mitochondrial genes limits the risk of amplifying nuclear pseudogene copies of mitochondrial DNA, aka numts, which have proven to be a real problem for phylogeny and molecular taxonomy (Song et al., 2008; Buhay, 2009; Hlaing et al., 2009; Hazkani-Covo, Zeller & Martin, 2010; Leite, 2012; Andújar et al., 2021; Graham, Gillespie & Krehenwinkel, 2021). However, as only five mitogenomes of Geoplanidae were available before this study, there remains a considerable amount of sequencing and documentation of additional taxa yet to be completed. We describe here for the first time the mitochondrial genomes of three already known species of Bipaliinae, namely B. vagum, B. adventitium and D. multilineatum, and we provide a map for B. kewense which was briefly reported without a map (Gastineau et al., 2019).

In this paper, for the species previously referred as Diversibipalium ‘black’ (Justine et al., 2018), additional material was obtained and we were able to prepare a formal description that includes morphology, histology and molecular phylogenies based on complete mitochondrial genome and nuclear ribosomal genes, and to finally assign this species, on the basis of its anatomy, to the genus Humbertium Ogren & Sluys, 2001.

For the second species, Diversibipalium ‘blue’, the low number of samples precluded histological investigation, but enough DNA was obtained for us to perform next generation sequencing and retrieve its full mitochondrial genome and nuclear ribosomal genes. We describe it as a new species of the genus Diversibipalium, which by definition does not imply any phylogenetic relationships except its appurtenance to the subfamily Bipaliinae. However, all phylogenies inferred from our molecular results positioned this species as the sister-group to all other Bipaliinae, thus suggesting that the species belongs to a different genus yet to be described.

Materials and Methods

Collection of specimens

New specimens of Diversibipalium “black” were provided in the context of a Citizen Science initiative (Justine et al., 2014, 2015, 2018, 2020a, 2020c, 2021) by Mrs Geneviève Rolland-Martinez, from her garden in Billère, Department of Pyrénées-Atlantiques, France. One of us (ER) collected numerous specimens in a private home garden located in Casier, Province of Treviso, Italy. In both cases, living specimens were sent by post to PG for photography and JLJ and RG for molecular work. Specimens were deposited in the Muséum National d’Histoire Naturelle in Paris, France (MNHN). A specimen of B. adventitium was collected in ethanol in Montréal, Québec, Canada on 27 May 2018 by Thomas Théry and deposited as MNHN JL328 (Justine et al., 2019). A specimen of B. vagum was collected in ethanol in Morne Vert, Martinique, Caribbean, on 19 November 2015 by Mathieu Coulis and deposited as MNHN JL307 (Justine et al., 2018).

Histology

The specimens were killed in boiling water and fixed in 95% pure ethanol for molecular studies. Specimens for histology were processed and stained by methods provided by Winsor & Sluys (2018). As the specimens were brittle, they were gradually hydrated through a series of descending ethanol solutions to water and softened in Sandison’s fluid until flexible (weeks). Specimens were then divided into anterior, pre-pharyngeal and posterior pieces, rinsed in water, and dehydrated in an ascending ethanol series to 95% ethanol-5% phenol, transferred to Supercedrol® (G.T. Gurr Ltd, London, UK), and infiltrated and embedded in Paraplast® paraffin wax, melting point 56 °C (McCormick Scientific). Tissue blocks were sectioned at 7 µm using a Leitz 1,212 rotary microtome, mounted on glass slides with Mayer’s albumen adhesive, stained by Cason’s modification of Mallory’s trichrome along with control sections, and mounted in Entellan® New (Merck-Millipore, Burlington, MA, USA).

Calculation of the Cutaneous Muscular Index (CMI) follows that of Froehlich (1955) and calculation of the Parenchymal Muscular Index (PMI) that of Winsor (1983).

Cox1 and LSU sequences obtained by Sanger method

Cox1 and LSU sequences were obtained by Sanger sequencing as detailed in Justine et al. (2018).

We built a tree and evaluated distances between all partial cox1 sequences available for the species previously referred to as Diversibipalium “black” from three localities. All alignment and analyses were conducted with MEGA7 (Kumar, Stecher & Tamura, 2016). After choosing the best model, which was the Hasegawa-Kishino-Yano model, an ML tree was constructed (Hasegawa, Kishino & Yano, 1985). A neighbour-joining (NJ) tree was constructed for comparison. Distances were analysed following routine methods (Justine et al., 2018).

Next generation sequencing and phylogeny

Samples of tissues conserved in ethanol 70% were sent to the Beijing Genomics Institute (BGI-Shenzhen, Shenzhen, China), which performed DNA extraction, library preparation and sequencing on a DNBSEQ platform. For each sample, a total of ca. 60 million clean 100 bp paired-end reads were obtained and assembled using SPAdes 3.14.0 (Bankevich et al., 2012) with a k-mer of 85. The contigs corresponding to mitogenomes were verified using Consed (Gordon, Abajian & Green, 1998). Genes were identified with the help of MITOS (Bernt et al., 2013), but also required manual curation on several occasions. rRNAs were obtained by alignments with reference sequences from O. nungara and B. kewense, and tRNAs were found using MITOS. In some cases, tRNAs were also checked with ARWEN command line using the -gcflatworm option (Laslett & Canbäck, 2008). All genomic maps were drawn using OGDRAW (Lohse et al., 2013). LOGOs were obtained from WebLogo3 online (Crooks et al., 2004). When needed, alignments were printed out using GenDoc (Nicholas, Nicholas & Deerfield, 1997).

SSU and LSU sequences were retrieved from the contig files obtained after assembly, by basic data mining using blastn command line and earlier references obtained by PCR as a database (Boratyn et al., 2012).

Four separate phylogenies were constructed, based on the partial nuclear ribosomal small subunit gene (SSU), the partial nuclear ribosomal large subunit gene (LSU), the concatenated amino-acid sequences of all mitochondrial proteins, and concatenated cox1, SSU and LSU genes. For SSU, 14 different sequences were used, and 20 for LSU with, in both cases the Geoplaninae O. nungara and A. expatria as outgroups. For the mitochondrial protein phylogeny, sequences obtained from 19 organisms were used, but here the outgroup was Prosthiostomum siphunculus Delle Chiaje, 1822 (Polycladida). The three-gene phylogeny (cox1, SSU, LSU) was performed on the same species as those included in the mitochondrial protein phylogeny, minus those for which SSU or LSU data were missing, plus the species Novibipalium venosum (Kaburaki, 1922) and Bipalium nobile, for which cox1 sequences HM346599 and MG436936 were used, respectively. Otherwise, all cox1 sequences were derived from whole mitogenomes, and SSU and LSU sequences correspond to those included in their respective trees. In total, 19 organisms were included in the three-gene phylogeny. The single genes and concatenated sequences were aligned using MAFFT 7 (Katoh & Standley, 2013) with the -auto function. For both concatenated datasets, the resulting alignments were trimmed by trimAl (Capella-Gutiérrez, Silla-Martínez & Gabaldón, 2009) with the -automated1 function. The final sizes of the trimmed alignments were 2,587 AA for the mitochondrial protein dataset and 3,447 bp for the three-gene dataset. For the SSU, LSU and three-gene phylogenies, the evolution model was GTR+I+G, chosen according to jModelTest2 (Darriba et al., 2012), while for the mitochondrial protein phylogeny it was mtART+I+G, chosen ad hoc as a model for mitochondrial protein coding genes of invertebrates (Abascal, Posada & Zardoya, 2007). Maximum likelihood (ML) phylogenies were all conducted using RaxML 8.0 (Stamatakis, 2014), with the best tree out of 100 being computed for 1,000 bootstrap replicates. Bayesian inference (BI) phylogenies were conducted on MrBayes 3.2.7 (Ronquist et al., 2012) using the default parameters, on alignments transformed into the nexus format by ALTER (Glez-Peña et al., 2010). Due to the absence of the mtART model in MrBayes 3.2.7, no BI phylogeny was performed on the concatenated mitochondrial protein sequences. The average standard deviations of split frequencies attained by MrBayes at the end of the run were 0.005317, 0.003415 and 0.001453 for the SSU, LSU and three-gene phylogenies, respectively.

Detection of alien DNA

For all the samples sequenced in this study, data mining was performed on the contigs obtained after assembly to find potential traces of alien DNA, using blastn command line (Boratyn et al., 2012) and a database consisting of SSU sequences from Eisenia fetida Savigny, 1826 (EF534709), Helix aspersa Müller, 1774 (MK919694) and Schistocerca pallens Thunberg, 1815 (KM853186).

Compliance with the international commission on zoological nomenclature

The electronic version of this article in Portable Document Format (PDF) will represent a published work according to the International Commission on Zoological Nomenclature (ICZN), and hence the new names contained in the electronic version are effectively published under that Code from the electronic edition alone. This published work and the nomenclatural acts it contains have been registered in ZooBank, the online registration system for the ICZN. The ZooBank LSIDs (Life Science Identifiers) can be resolved and the associated information viewed through any standard web browser by appending the LSID to the prefix http://zoobank.org/. The LSID for this publication is: urn:lsid:zoobank.org:pub:27A4D685-9042-40C2-A40A-89FF8BCC489B. The online version of this work is archived and available from the following digital repositories: PeerJ, PubMed Central SCIE and CLOCKSS.

Results

Description of Humbertium covidum

Taxonomy

Order Tricladida Lang, 1884 (Lang, 1884)

Suborder Continenticola Carranza, Littlewood, Clough, Ruiz-Trillo, Baguña & Riutort, 1998 (Carranza et al., 1998)

Family Geoplanidae Stimpson, 1857 (Stimpson, 1857)

Subfamily Bipaliinae von Graff, 1896 (von Graff, 1896)

Genus Humbertium Ogren & Sluys, 2001 (Ogren & Sluys, 2001)

Humbertium covidum n. sp.

urn:lsid:zoobank.org:act:3847E9FE-463B-4FDB-A164-88765A52D65A

Synonym: Diversibipalium “black” of Justine et al. (2018)

Type-locality: Garden in Casier, county of Casier, Province of Treviso, Region of Veneto, Italy. Coordinates: E 12.289391, N 45.639459. Collected by Enrico Ruzzier on 30th September 2019.

Type-material: Holotype MNHN JL351B (36 microslides anterior LSS, pre-pharyngeal TS and posterior LSS in a single block) and Paratypes MNHN JL351A (34 microslides TS anterior half and LSS posterior portion in a single block); 14 Paratypes (MNHN JL351C-G and JL351Q-Y) retained whole.

Additional material and localities: MNHN JL090, six specimens, domestic garden in Saint-Pée-sur-Nivelle, Department of Pyrénées Atlantiques, France, collected 12 November 2013; MNHN JL343, one specimen, domestic garden in Billère, Department of Pyrénées Atlantiques, France, collected 14 May 2019.

Behaviour and habitat: In Casier, Italy, the species was the only flatworm found; numerous specimens were swarming, and the species was active in the earliest hours of the morning, not during late evening or at night. The two records from France were from gardens where Bipalium kewense was also found.

Molecular information: MNHN JL090 from Saint-Pée-sur-Nivelle: partial cox1 sequence from Sanger sequencing MG655588 (Justine et al., 2018); partial LSU from NGS, MZ520989 (this paper); partial SSU from NGS, MZ520996 (this paper); complete mitogenome MZ561471 (this paper). MNHN JL343 from Billère: partial cox1 sequence from Sanger sequencing, MZ622153 (this paper). MNHN JL 351 from Casier, partial cox1 sequence, five replicates from Sanger sequencing MZ622148–MZ622152 (this paper); partial LSU, three replicates from Sanger sequencing MZ647546–MZ647548 (this paper); partial LSU from NGS MZ520988 (this paper); partial SSU from NGS MZ520995 (this paper); complete mitogenome MZ561472 (this paper). See File S1 for details.

Etymology: The specific name covidum was chosen as homage to the numerous casualties worldwide of the COVID-19 pandemic. Furthermore, a large part of this study was written during the lockdowns.

Similarity of cox1 sequences from various populations

For the specimens from Casier, Italy, we had six cox1 sequences, including five from Sanger sequencing (MNHN JL351H, J, K, L, M) and one from the NGS mitogenome. In addition, we had the cox1 sequence of one specimen from Billère, Pyrénées Atlantiques (MNHN JL343) and the cox1 sequences of two specimens MNHN JL090 from Saint-Pée-sur-Nivelle, Pyrénées Atlantiques, mentioned in our 2018 paper (Justine et al., 2018) and described as Diversibipalium sp. “black”, one from Sanger sequencing and one from the NGS mitogenome. The ML and NJ trees (Fig. 1) built from these nine sequences, and one sequence of B. kewense as the outgroup were identical and showed that sequences were separated into two clades: one clade included all sequences from France (both from Billère and Saint-Pée-sur-Nivelle) and all these sequences were identical; one clade included all sequences from Italy and all these sequences were identical. The differences between two clades involved 10 positions out of 387, i.e., the distance based on partial cox1 sequences was 2.58%.

Figure 1 Humbertium covidum n. sp. from two populations, tree based on cox1 sequences.

The evolutionary history was inferred using the Maximum Likelihood and the Neighbour-Joining methods; there was a total of 387 positions in the final dataset. All partial cox1 sequences from Italy (six specimens) were identical, as were the three sequences from France, from two localities. Sequences from France and Italy differed by 2.58%. Bootstrap values: above branches, ML; below branches, NJ.

Diagnosis

Specimens of Humbertium with reniform-shaped headplate, with dark brown to black dorsal ground colour, without stripes or other ornamentation, ventral surface light grey–greyish brown with paler creeping sole; eyes in a triple row around anterior headplate, present dorso-laterally on headplate, ventrally behind the lappets, continuing along the sides of the body in a staggered row posteriorly; pharynx plicate; testes ventral, extending from behind ovaries to pharynx; vas deferens enter penis bulb separately; penis bulb small, strongly muscularised; penis almost horizontal, elongate, tapered; male atrium almost horizontal then steeply inclined ventrally; female genital canal almost vertical, in two parts with shell glands opening into the proximal canal; ovovitelline ducts ascend dorsally before the gonopore to enter the proximal female glandular canal antero-dorsally; the male and female efferent ducts are contained within a muscular genital pad through which the female canal opens to the left and slightly dorsal to the male canal, both entering the common genital canal slightly posterior and above the gonopore. A viscid gland is present in the genital pad anterior to the male efferent duct. The efferent canals open into a narrow horizontal highly glandular common genital canal. The common canal opens into the common atrium, in which the gonopore is centrally placed ventrally.

Morphology

Photographs of specimens are presented for live specimens from Italy (Figs. 2–5) and Billère in France (Figs. 6–10) and preserved specimens from Saint-Pée-sur-Nivelle (Figs. 11, 12). Headplate reniform with rounded non-recurved lappets, with width of headplate in living specimens about 1.3 times the maximum body width, and headplate length to width ratio 1:1.6–2.7 (measured from scaled drawings of photographs of living specimens, Fig 19 Justine et al., 2018), and 0.8 times the maximum body width in preserved specimens. Living specimens attain a length of 20–25 mm, and preserved specimens 9–20 mm in length, with the body width:length ratio 1:4.5–1:5.7. Dorsal ground colour dark brown to black, with no evidence of dorsal stripes or bands on body or headplate (Figs. 2, 3, 5, 6). Ventral surface light grey to greyish-brown colour with pale grey creeping sole (Figs. 3, 4). Dimensions of preserved sexual specimens are provided in Table 2 and Figs. 11, 12.

Figure 2 Humbertium covidum n. sp. from Italy, alive.

General dorsal aspect. Photo by Pierre Gros.

Figure 3 Humbertium covidum n. sp. from Italy, alive.

Lateral view showing locomotion and slime trail. Photo by Pierre Gros.

Figure 4 Humbertium covidum n. sp. from Italy, alive.

Individual with raised anterior end showing ventral surface. Photo by Pierre Gros.

Figure 5 Humbertium covidum n. sp. from Italy, alive.

Ventral surface with typical headplate shape. Photo by Pierre Gros.

Figure 6 Humbertium covidum n. sp. from Billère, France, alive.

General dorsal aspect. Photo by Pierre Gros.

Figure 7 Humbertium covidum n. sp. from Billère, France, alive.

Lateral aspect. Photo by Pierre Gros.

Figure 8 Humbertium covidum n. sp. from Billère, France, alive.

Lateral aspect showing extended papillae on headplate. Photo by Pierre Gros.

Figure 9 Humbertium covidum n. sp. from Billère, France, alive.

Individual with raised anterior end. Photo by Pierre Gros.

Figure 10 Humbertium covidum n. sp. from Billère, France, alive.

The flatworm seems to threaten a snail (unidentified species). Photo by Pierre Gros.

Figure 11 Humbertium covidum n. sp. from Saint-Pée-sur-Nivelle, France, preserved.

Specimen MNHN JL090, preserved specimen, dorsal aspect. Showing the partly protruded pharynx. Photo by Jean-Lou Justine. Reproduced from Figure 20 of Justine et al. (2018).

Figure 12 Humbertium covidum n. sp. from Saint-Pée-sur-Nivelle, France, preserved.

Specimen MNHN JL090. Preserved specimen, ventral aspect. The ventral ground colour is grey, with the creeping sole a lighter tone. The pharynx is slightly protruded from the mouth, and the gonopore is evident as a small transverse white slit on the creeping sole some 2 mm below to the mouth. Scale is in mm. Photo by Jean-Lou Justine. Reproduced from Figure 21 of Justine et al. (2018).

Table 2 Humbertium covidum n. sp.: dimensions of specimens examined.

Fixed specimen dimensions	Holotype
JL 351A	Paratype
JL 351A	Paratype
JL 351A	Paratype
JL 351A	Paratype
JL 351A	Voucher
JL 090	
Geographic origin	Italy	Italy	Italy	Italy	Italy	France	
Length (mm)	13.3	12.0	13.2	12.8	9.0	20.0	
Width of headplate (mm)	2.0	NM	2.0	2.0	2.0	NM	
Width at mouth (mm)	2.5	2.1	2.5	2.5	2.4	3.2	
Ratio width headplate to body width	0.8:1	–	0.8:1	0.8:1	0.8:1	–	
Ratio body width to length	1:5.3	1:5.7	1:5.3	1:5.1	1:4.5	1:6.3	
Mouth (mm)	5.0 (37.6)	5.6 (46.7)	6.5 (49.2)	6.0 (46.9)	6.0 (66.7)	6.0 (30.0)	
Gonopore (mm)	7.1 (53.4)	7.5 (62.5)	8.2 (62.1)	7.8 (60.9)	7.4 (82.2)	7.8 (39.0)	
Distance mouth-gonopore (mm)	2.1 (15.8)	1.9 (19)	1.7 (12.9)	1.8 (14.1)	1.4 (15.6)	1.8 (9)	
Body Height µm	1,157	741					
Creeping sole width µm (% body width)	1,157 (30.8%)	647 (21.6%)					
CMI	2%	3.2%					
PMI	8.4%	15%					
Pharynx type	Collar	Bell-Collar					
Pharyngeal pouch length (% body length)	1 068 µm (8%)	1 287 µm (10%)					
Position of mouth in pharyngeal pouch	623 µm (58.5%)	624 µm (48.5%)					
Distance between posteriad pharyngeal pouch and anteriad penis bulb	250 µm	590 µm					
Note:

Positions of body apertures are measured from the anterior tip. Figures in parentheses are the position of the aperture expressed as a percentage of body length. NM: not measured.

Internal anatomy

Body wall and musculature

These characteristics are shown in Figs. 13, 14. The epithelium is thicker dorsally (28–32 µm) than ventrally (12–21 µm). Three types of rhabdoids are present: large xanthophil chondrocytes measuring 23.8 × 5.6 µm to 30.8 × 4.2 µm (length × width) predominate over the dorsum to the marginal zone, and xanthophil rhammites measuring 21.0 × 1.4 µm–25.2 × 2.8 µm (length × width) also cover the same area but are less numerous. Both the chondrocytes and rhammites project slightly above the epithelium. Micro-rhabdites (stäbschen) 2.8–4.2 µm × 0.7 µm (length × width) are present in the ventral epithelium, mainly either side of the creeping sole. Of the epidermal secretions, xanthophil secretions predominate over the dorsum to the marginal zone, with erythrophil and cyanophil granular secretions relatively sparse except over the creeping sole, with a small concentration of erythrophil secretions either side of the slight central protuberance on the creeping sole. Epidermal secretions on the headplate reflected those of the rest of the body. All epidermal secretions are derived from mesenchymal secretory cells. There is no evidence of a glandular margin. Fine black granular pigment is sparsely scattered throughout the dorsal mesenchyme, though it was noted that Sanderson’s fluid appeared to elute some black dorsal pigment. The ciliated creeping sole is 21.6–30.8% of the body width and is slightly protuberant centrally and bears an insunk ciliated epithelium.

Figure 13 Anatomy of Humbertium covidum n. sp., pre-pharyngeal region.

Holotype, specimen MNHN JL351B. Pre-pharyngeal region, transverse section. Arrows indicate the extent of the creeping sole. Photo by Leigh Winsor.

Figure 14 Anatomy of Humbertium covidum n. sp, ventral longitudinal muscular plate.

Holotype, specimen MNHN JL351B. Lateral body showing ventral longitudinal muscular plate. Photo by Leigh Winsor.

Cutaneous musculature is tripartite and very weakly developed, comprising circular muscle represented by a single fibre, single decussate diagonal fibres, and longitudinal muscles in small bundles of 2–3 fibres each, CMI 2–3%.

Parenchymal musculature consists of a strong ventral plate of longitudinal muscles extending laterally to the mid-lateral region and divided into uneven bundles of 4–10 fibres by dorsoventral muscles, with weak supraneural and dorsal parenchymal longitudinal muscles present as single fibres. Strong dorsal transverse muscles and weak supraintestinal and dorsoventral muscles are present. PMI 8.4–15% of which the dorsal transverse muscles contribute the greater amount.

Alimentary system

The pharynx is collar-form (Fig. 15), with the dorsal insertion in the posterior third of the pharyngeal pouch and posterior to the mouth, and the ventral insertion anterior to the mouth. The outer pharyngeal musculature comprises an ectal single fibre of longitudinal muscle underlain by circular muscles and an ental layer of longitudinal muscles. The inner musculature consists of a single longitudinal muscle fibre underlying the insunk epithelium, underlain by sheaths of circular and longitudinal muscles (derived planariid type). Radial muscles, erythrophil, xanthophil and cyanophil secretory ducts make up the mid-pharyngeal wall. The pharyngeal pouch is 1,068–1,342 µm long, representing 8–10% of the total body length. The mouth is situated in the approximate mid-ventral region of the pouch. Oesophagus absent.

Figure 15 Anatomy of Humbertium covidum n. sp., pharynx.

Holotype, specimen MNHN JL351B. Pharynx, sagittal section. Photo by Leigh Winsor.

Sensory organs

The sensorial zone contours the anterolateral sub-margin of the headplate and consists of flat tooth-like aciliate papillae about 55 µm high and 38 µm wide separated from each other by a groove of 10 µm, with about 20 papillae per millimetre. Ciliated pits about 20 µm deep and 4 µm wide open just below the lips of the papillae.

Eyes are present as a triple row contouring the anterior margin of the headplate, with extension dorso-laterally, and ventrally behind the lappets, then continuing posteriorly along the body sides in a staggered row (Fig. 16). The eyes are pigment cup ocelli of similar shape and size, about 16 µm in diameter, with two retinal clubs per ocellus.

Figure 16 Morphology of Humbertium covidum n. sp., eye pattern.

Paratype JL 351C. Headplate showing the dorsal and ventral eye patterns in a cleared specimen. The headplate is curled ventrad. Drawing by Leigh Winsor.

Reproductive organs

The ovaries are spheroidal, 150 µm in diameter, located almost a millimetre behind the anterior margin of the headplate and are half embedded in the ventral nerve cords.

The testes are ventral, round to ovoid in shape about 300 µm high and 220 µm in diameter and extend uniserially from behind the ovaries posteriorly to the pharynx. They open towards the lower testicular pole via short sperm ductules into the vasa deferentia. The vasa deferentia of both sectioned specimens of Humbertium covidum contained mature spermatozoa. In both specimens, the testes are at different stages of maturity with those nearest the copulatory organs containing mature spermatozoa.

The copulatory organs (Figs. 17, 18) lie 250–590 µm behind the pharyngeal pouch. The male organ rises 20° dorsad from the horizontal and the male atrium dips steeply 50° ventrad, with the female organ almost vertically positioned (10° from the vertical towards the posterior).

Figure 17 Anatomy of Humbertium covidum n. sp., composite drawing of copulatory organs.

Holotype, specimen MNHN JL351B. Composite reconstruction of the copulatory organs, sagittal view. The dashed line in the common atrium indicates the extent of the glandular mesenchyme forming the common genital canal. Anterior: left. Drawing by Leigh Winsor.

Figure 18 Anatomy of Humbertium covidum n. sp., level of gonopore.

Holotype, specimen MNHN JL351B. Copulatory organs at the level of the gonopore, with the female glandular canal entering the common genital canal at the point where it communicates with the common atrium. Anterior: left. Photo by Leigh Winsor.

Male organs

The protrusible penis comprises a small but highly muscular bulb, with an elongate, ventrally curved, and tapered finger-like papilla opening towards the left-hand side, and filling most of the conical male atrium. The lumen of the seminal (prostatic) vesicle is 50–60 µm in diameter and lined by a cuboidal secretory epithelium receiving fine granular erythrophil secretions from erythrophil mesenchymal glands surrounding the bulb. This epithelium grades into a tall voluminous nucleate columnar epithelium penetrated by the expanded terminal ducts of mesenchymal erythrophil glands external to the bulb, discharging secretions into the proximal ejaculatory duct. At about the point where the penis bends towards the gonopore, the lining of the ejaculatory duct transitions to a cuboidal epithelium of the distal ejaculatory duct with a reduction in secretions and height, and from there grades to the flat nucleate-facing epithelium of the distal penis papilla.

The dorsal half of male atrium is lined by a low-facing epithelium and the ventral half lined by a low nucleate columnar epithelium that also covers the proximal external penis. Distally the atrium is lined by a low-facing epithelium. An inner strong sheath of circular muscles and an external sheath of longitudinal muscles underlie the atrial epithelium.

Musculature of the penis bulb consists of a strong outer sheath of broad bands of longitudinal muscles between which oblique muscles are interwoven. The ejaculatory duct is surrounded by a strong inner layer of circular muscles underlain by mixed longitudinal and circular muscles, with a sheath of circular muscles underlying the outer penial epithelium.

The vasa deferentia, lined by a cuboidal epithelium, lie lateral to and on the same level as the ovovitelline ducts, and just below the testes with which they communicate via a short sperm ductule. Passing posteriorly, they continue to the level of the penis bulb where they gently rise, expand to form spermiducal vesicles and recurve, piercing the anteriad penis bulb to separately open into the seminal vesicle.

Female organs

The glandular canal is aligned almost at right angles to the ventral surface, is about 370 µm in length, and is divided into two distinct parts-the proximal (dorsad) section, and the distal (ventrad) section: the proximal two-thirds of the glandular canal is thistle-shaped with a maximum diameter of around 220 µm, with a distinct constriction before the centrally invaginated flared blind end where the ovovitelline ducts debouche (could be termed the seminal receptacle). The proximal glandular canal is lined by a tall columnar epithelium with basal nuclei. Secretory ducts, from erythrophil (shell-glands) and cyanophil glands in the surrounding mesenchyme, pierce the epithelium to discharge their contents into the lumen. The fine granular secretions from both types of glands condense within the epithelium and are secreted as membrane-bound masses into the glandular canal. The distal third of the glandular canal, lined by a non-secretory ciliated columnar epithelium, narrows to 70 µm then tapers to 16 µm to discharge into the common genital canal. Underlying the epithelium of the glandular canal is a layer of circular muscles external to which are longitudinal muscles, the whole being invested in a weak muscularis.

The ovovitelline ducts, lined by a ciliated cuboidal epithelium with a circular muscularis, emerge from the lower poles of the ovaries, ascend slightly to pass posteriorly along the lateral margins of the nerve cords. The ovovitelline ducts turn dorsally before the gonopore (holotype JL351B; in the paratype JL351A they turn dorsally about level with the posterior lip of the gonopore some 100 µm posteriad to that figured for the holotype), rise and enter the female glandular canal antero-dorsally, exhibiting the proflexed condition.

Common genital canal and common atrium

In the mesenchyme below the male and female organs in the left body wall lies a crescentic band of densely aggregated erythrophil and basiphil glands (Fig. 19). Moving from the left to the right through the mesenchyme, a crescentic split develops in the body wall ventral to the glandular mesenchyme that eventually enlarges to become the common atrium (Fig. 20). At about the same point, an elongate fissure develops horizontally along the mid-band of the glandular mesenchyme, in what becomes the common genital canal on the dorsal side of the fissure. For about 60 µm across the body, the wall of the mesenchyme separates the common genital canal from the common atrium (Fig. 21). The common genital canal is lined by a highly glandular insunk epithelium, richly endowed with granular lightly erythrophil secretions that are secreted as packets of erythrophil granules, alternating with cyanophil strand secretions, typical of the secretory elements related to cocoon formation. The common atrium is lined by an insunk columnar epithelium, through which amorphous cyanophil secretions are discharged. Commensurate with the appearance of the gonopore, the mesenchymal wall thins to form a residual flap around what becomes a single common atrium. Numerous erythrophil glands discharge their granular secretions into the crease formed between the residual flap and genital pad.

Figure 19 Anatomy of Humbertium covidum n. sp., putative common genital canal.

Paratype, specimen MNHN JL351C. Glandular mesenchyme of the putative common genital canal on the left side of the body. Anterior: left. Photo by Leigh Winsor.

Figure 20 Anatomy of Humbertium covidum n. sp., common genital canal.

Paratype, specimen MNHN JL351C. The beginning of the slit-like common genital canal. Anterior: left. Photo by Leigh Winsor.

Figure 21 Anatomy of Humbertium covidum n. sp., male atrium.

Paratype, specimen MNHN JL351C. The point where the male atrium is about to open into the common genital canal which has not yet opened into the common atrium. Anterior: left. Photo by Leigh Winsor.

Genital pad and viscid gland

The genital pad comprises strong interwoven circular and oblique muscles covered by the same epithelium as the common genital canal. In the anteriad pad is situated an ovoid-shaped viscid gland (Figs.22, 23), 280 µm high and 120–200 µm in diameter, with a duct 160–200 µm long and 20–70 µm in diameter that the opens into the common genital canal. The epithelium of the viscid gland is predominantly charged with finely granular cyanophil secretions, alternating with packets of amorphous dark erythrophil secretions both derived from glands in the surrounding mesenchyme. The secretions discharged into the lumen combine to form thin dark basiphil strands.

Figure 22 Anatomy of Humbertium covidum n. sp., viscid gland.

Holotype, specimen MNHN JL351B. The viscid gland at the anteriad end of the genital pad below the male organs. Anterior: left. Photo by Leigh Winsor.

Figure 23 Anatomy of Humbertium covidum n. sp., viscid gland and erythrophil glands.

Holotype, specimen MNHN JL351B. The glandular duct of the viscid gland, and erythrophil glands in the atrial crease. Anterior: left. Photo by Leigh Winsor.

Vitellaria are sparse and lie between diverticula of the gut. A genito-intestinal duct is absent.

Additional comments

Fixation

Stain uptake by the tissue sections of both specimens was suboptimal, in part due to the fixation in 95% ethanol resulting in pronounced tissue vacuolation, and possibly partly due to the prolonged post-fixation treatment in Sandison’s fluid. The control tissue sections included with the slides of Humbertium covidum verified that the Mallory stain worked perfectly on formaldehyde-fixed tissue.

Pathology

The larva of a nematode was present in the creeping sole of the holotype.

Video file

A short video file of a living specimen is available as File S2.

Occurrences

The species was recorded in 2013 from a single garden in Saint-Pée-sur-Nivelle (Department of Pyrénées Atlantiques, France) in which B. kewense was also present. According to the owner, the species was present for years in the garden and was still present in 2017. It was then found in a garden in Billère, in the same Department, ca. 100 km from the first location; this garden was also heavily infested with B. kewense (Justine et al., 2018). Finally, one of us found in 2019 an abundant population in Casier, Province of Treviso, Italy. In 2019, an intensive campaign on Twitter in various European languages asking for additional reports, did not provide any additional information. In the discussion, we report possible other occurrences in various countries.

Description of Diversibipalium mayottensis n. sp.

Taxonomy

Order Tricladida Lang, 1884 (Lang, 1884)

Suborder Continenticola Carranza, Littlewood, Clough, Ruiz-Trillo, Baguña & Riutort, 1998 (Carranza et al., 1998)

Family Geoplanidae Stimpson, 1857 (Stimpson, 1857)

Subfamily Bipaliinae von Graff, 1896 (von Graff, 1896)

Genus Diversibipalium Kawakatsu et al., 2002 (Kawakatsu et al., 2002)

Diversibipalium mayottensis n. sp.

urn:lsid:zoobank.org:act:B59FEE8E-70FD-4DEC-B839-554C351701F8

Synonym: Diversibipalium “blue” of Justine et al. (2018).

Type-locality: Ouangani, Mayotte.

Additional localities: Mtsamboro and Mamoudzou, Mayotte.

Type-material: Holotype MNHN JL282, Cascade du Mont Meoni ouaj Coconi, Commune of Ouangani, Mayotte; Coordinates: W 45.12936111, S 12.83522222; Collected on 30 April 2015; Photographed live (Figs. 24–27); length of preserved specimen 15 mm; cox1 sequence MG655598. Paratypes: MNHN JL280, Dziani, Commune of Mtsamboro, Mayotte; Coordinates: W 45.08091667, S 12.71208333, 29 April 2015; one specimen, head not visible; length of preserved specimen 7 mm; cox1 sequence MG655596. MNHN JL281, Dziani, Commune of Mtsamboro, Mayotte; Coordinates: W 45.08758333, S 12.69638889, 29 April 2015; specimen JL281A, length preserved 15 mm, first slightly damaged for Sanger sequencing, later almost completely destroyed for NGS sequencing (only head retained); JL281B, 5 mm; JL281C, length preserved 9 mm; cox1 sequence MG655597 (based on three identical replicates). MNHN JL283, Convalescence, Commune of Mamoudzou, Mayotte; Coordinates: W 45.18963889, S 12.76891667, 4 May 2015; one specimen, head not visible, preserved 20 mm, alive ca. 30 mm; not sequenced. MNHN JL284, Îlot Mtsamboro, Commune of Mtsamboro, Mayotte; Coordinates: W 45.02769444, S 12.64247222, 5 May 2015; one specimen, head visible, tail damaged, length preserved 12 mm, cox1 sequence MG655599. All specimens collected by Laurent Charles. See also File S1.

Figure 24 Diversibipalium mayottensis n. sp, alive.

Specimen MNHN JL282 from Mayotte, Indian Ocean, dorsal aspect. The headplate of this small planarian is a rusty-brown colour that extends to some irregular patches on the ‘neck.’ The dorsal ground colour is an iridescent blue–green (‘dark turquoise glitter’). Photo by Laurent Charles. Reproduced from Figure 23 in Justine et al. (2018).

Figure 25 Diversibipalium mayottensis n. sp, alive.

Specimen MNHN JL282 from Mayotte, Indian Ocean, dorsal aspect. Same specimen as in Fig. 24. Photo by Laurent Charles. Reproduced from Figure 24 in Justine et al. (2018).

Figure 26 Diversibipalium mayottensis n. sp, alive regenerating specimen.

Dorsal aspect of a regenerating specimen with a damaged anterior end. Specimen MNHN JL280. Under appropriate lighting, the colour of the specimen takes on a beautiful, almost metallic green colour. The iridescence and blue–green colour are lost on fixation, leaving the specimen a dark brown. Photo by Laurent Charles. Reproduced from Figure 25 in Justine et al. (2018).

. Figure 27 Diversibipalium mayottensis n. sp, alive regenerating specimen.

Dorsal aspect of a regenerating specimen with a damaged anterior end. Specimen MNHN JL280. A small portion of the brown-pigmented ventral surface with the median pale creeping sole can be seen. Photo by Laurent Charles. Reproduced from Figure 26 in Justine et al. (2018).

Behaviour and habitat: In Mayotte, all specimens were collected during the day, under dead wood or leaves, as part of a terrestrial mollusc program. No collection was attempted during the night. All localities were in a slightly degraded natural environment, with little human presence. No research was done to know whether the species was found in gardens, but no citizen science record was received that would suggest this is the case.

Molecular information: All partial cox1 sequences from 6 specimens listed above were identical; see Fig. 2 in Justine et al. (2018). One specimen (MNHN JL281A) used for NGS sequencing, providing sequences for SSU (MZ520997), LSU (MZ520986) and complete mitogenome (MZ561470).

Etymology: The specific name mayottensis refers to the type-locality.

Attribution of the species to Diversibipalium

The genus Diversibipalium Kawakatsu et al., 2002 is a collective group created to temporarily accommodate species whose anatomy of the copulatory apparatus is still unknown (Kawakatsu et al., 2002) and it is therefore logical that we attribute the new species to this genus.

Diagnosis

Specimens of Diversibipalium with a rusty-brown coloured club-shaped headplate, with iridescent blue green dorsal ground colour in life, dark brown colour when preserved, with the suggestion of a fine white median dorsal stripe; ventral surface light brown with white to pale green coloured creeping sole. The mouth is present in the anterior second fifth of the body, and gonopore in the fourth body fifth.

Morphology

The specimen has the overall morphology of a typical bipaliine, with the headplate of the living specimen is a rusty-brown colour that extends to some irregular patches on the “neck” (Figs. 24, 25). The dorsal ground colour is an iridescent blue green (“dark turquoise glitter”) (Figs. 24–27), with a hint of a fine white median stripe, and the ventral surface a light brown colour, with the creeping sole white to pale green. The iridescence and blue-green colour are lost on fixation, leaving a dark brown ground colour. The posterior margins of the headplate are not recurved but rounded (reniform), giving the headplate a club-shape, with width of headplate in living specimens 1.1–1.3 times the maximum body width, and headplate length to width ratio 1:0.6–0.7 (relative dimensions taken from photographs of living specimens in Figs. 24, 25). The living specimens are up to about 45 mm in length.

A preserved sexual specimen (paratype JL281C), 9 mm long and 1 mm wide, had the mouth situated ventrally approximately 3.5 mm (39% of the body length) from the anterior end, and gonopore 3 mm (33% of the body length) posterior to the mouth. All specimens were used for molecular analysis with the exception of JL 283. In view of the very few specimens available, no specimen was used for histological methods.

Mitochondrial genomes

New mitogenomes for five species

The main characteristics of all mitogenomes obtained during this study are summarised in Table 3. The genomic maps are also presented for H. covidum JL351 (Fig. 28), H. covidum JL090 (Fig. 29), D. mayottensis JL281 (Fig. 30), B. vagum JL307 (Fig. 31), B. adventitium JL328 (Fig. 32) and D. multilineatum JL177 (Fig. 33). We also present the genomic map of B. kewense (Fig. 34). With the exception of D. multilineatum JL177, all mitogenomes seemed complete, and all are colinear concerning protein-coding and rRNA genes. The situation with tRNA is slightly different. The number of tRNAs found among the mitogenomes varies between 21 to 22. For example, it was impossible to find a tRNA-Thr for both specimens of H. covidum, while it is commonly found in the cluster of tRNA comprised between cob and rrnL in other species such as B. kewense, D. mayottensis or B. vagum. Also, B. adventitium singularizes itself from the others by the total lack of tRNA cluster in the aforementioned area. Instead, two of these tRNAs, tRNA-Asn and tRNA-Leu were found in an intergenic area between ND5 and ND6. In a similar situation to that explained below regarding the 16S gene, it should be noted that it was often difficult to detect tRNA among these specimens.

Figure 28 Mitogenome of Humbertium covidum n. sp.: genomic map of specimen MNHN JL351.

Specimen from the Italian population in Casier. The mitogenome is 15,540 bp long and contains 12 protein coding genes, two ribosomal RNA genes and 21 transfer RNA genes. The ND3 gene was found with a premature stop codon.

Figure 29 Mitogenome of Humbertium covidum n. sp.: genomic map of specimen MNHN JL090.

Specimen from the French population in Billère (Pyrénées-Atlantiques). The mitogenome is 15,524 bp long and contains 12 protein coding genes, two ribosomal RNA genes and 21 transfer RNA genes. The ND3 gene was found with a premature stop codon.

Figure 30 Mitogenome of Diversibipalium mayottensis n. sp.: genomic map of specimen JL281.

The mitogenome is 15,989 bp long and contains 12 protein coding genes, two ribosomal RNA genes and 22 transfer RNA genes.

Figure 31 Mitogenome of Bipalium vagum: genomic map of specimen JL307.

The mitogenome is 17,149 bp long and contains 12 protein coding genes, two ribosomal RNA genes and 22 transfer RNA genes. The genes cox3, atp6, ND1, ND4L have alternative start codon.

Figure 32 Mitogenome of Bipalium adventitium: genomic map of specimen JL328.

The mitogenome is 15,494 bp long and contains 12 protein coding genes, two ribosomal RNA genes and 21 transfer RNA genes. It was not possible to find a stop codon for the cob gene.

Figure 33 Mitogenome of Diversibipalium multilineatum: genomic map of specimen JL177.

The mitogenome is not complete. The size of the partial genome is 15,660 bp long and contains 12 protein coding genes, two ribosomal RNA genes and 21 transfer RNA genes. The genes ND2 and ND3 have alternative start codon.

Figure 34 Mitogenome of Bipalium kewense: genomic map of specimen JL184A.

The mitogenome is 15,666 bp long and contains 12 protein coding genes, two ribosomal RNA genes and 22 transfer RNA genes.

Table 3 Characteristics of mitogenomes of bipaliines.

Species	MNHN registration number	GenBank accession number	Size of the mitogenome	Early stop	Alternative start codon	
Humbertium covidum	JL351	MZ561472	15,540 bp	ND3	–	
Humbertium covidum	JL090	MZ561471	15,524 bp	ND3	–	
Diversibipalium mayottensis	JL281	MZ561470	15,989 bp	-	–	
Bipalium vagum	JL307	MZ561468	17,149 bp	-	cox3, atp6, ND1, ND4L	
Diversibipalium multilineatum	JL177	MZ561469	15,660 bp
(not complete)	-	ND2, ND3	
Bipalium adventitium	JL328	MZ561467	15,494 bp	cob	–	
Bipalium kewense	JL184A	MK455837	15,666 bp	–	–	

No putative ATP8 gene could be evidenced so far. Blastx analyses of all mitogenomes from Bipaliinae were done against a customized database that included the putative ATP8 amino-acid sequences of Stenostomum sthenum Borkott, 1970 (ARW59252) and Macrostomum lignano Ladurner, Schärer, Salvenmoser & Rieger 2005 (ARW59249) from Egger, Bachmann & Fromm (2017), and also the putative ORF neighbouring the ND2 gene of Girardia sp. (KP090061) and Phagocata gracilis Haldeman, 1840 (KP090060), considered by Ross et al. (2016) as putative highly divergent ATP8. All attempts failed to find any ATP8 candidate among the Bipaliinae.

Genomic comparison at the population level of H. covidum

Table 4 lists the protein-coding genes of H. covidum, and compares the sequences obtained from JL351 (from Italy) and JL090 (from France). All mitochondrial protein-coding genes were found to display polymorphisms, some of them being non-silent. A gene commonly used for molecular barcoding and phylogeny such as the cox1 gene showed 35 polymorphisms on 1,551 bp, which corresponds to a percentage of difference of 2.25%. This difference is interpreted as intraspecific. As a comparison, cox1 alignment between Dugesia japonica Ichikawa & Kawakatsu, 1964 and D. ryukyuensis Kawakatsu, 1976 showed a much larger difference of 17.91%. Similarly, B. kewense showed 16.93% differences with B. adventitium and 15.7% with B. vagum. Noticeable differences, which include SNPs and indels, were found in the 16S rRNA genes of the two specimens of H. covidum, as described below.

Table 4 Genetic differences between two populations (JL351, Italy vs JL090, France) of Humbertium covidum.

	atp6	cob	cox1	cox2	cox3	ND1	ND2	ND3	ND4	ND4L	ND5	ND6	
Polymorphic site (nucleotides)	19/669	36/1,110	35/1,551	20/747	20/786	19/897	17/870	6/352	27/1,407	8/291	46/1,599	15/492	
Percentage of differences (nucleotides)	2.84	3.24	2.25	2.68	2.54	2.19	1.95	1.70	1.92	2.75	2.88	3.05	
Polymorphic sites (amino-acids)	8/222	7/369	3/516	4/248	1/261	7/298	4/289	2/117	4/468	3/96	11/532	7/163	
Percentage of differences (amino-acids)	3.60	1.90	0.58	1.61	0.38	2.34	1.38	1.70	0.85	3.125	2.07	4.29	

The peculiar case of the 16S gene

As more mitogenomes of Bipaliinae have been sequenced, a recurrent issue has arisen. Systematically, tools such as MITOS and MITOS2 were unable to locate the exact position of the 16S gene. For example, when submitting the mitogenome of H. covidum JL351 to these software programmes, only a 563 bp fragment was recognised, meaning that a large subunit of the ribosome, which is smaller than the small subunit, is itself 726 bp long. To verify the putative position of the 16S, additional alignments were performed with the reference sequence from Schmidtea mediterranea Benazzi, Baguñà, Ballester, Puccinelli & Del Papa, 1975 (JX398125), which has the advantage of having been verified by RNAseq (Ross et al., 2016). With such a method, a putative gene of 1,063 bp was detected for H. covidum JL351. A similar problem arose with all other species. An alignment of the ‘complete’ 16S genes from all Bipaliinae is displayed as a LOGO and shown in Fig. 35. The portion that corresponds to the 563 bp fragment suggested by MITOS corresponds to the portion that starts around position 530, which delimitates the beginning of a more conserved portion of the gene. The alignments shown in Fig. 36 were obtained from the 16S genes of both specimens of H. covidum, and show where the polymorphisms and indels occurred. In the most conserved region, the start of which is indicated by a star, seven polymorphisms were found, while 12 polymorphisms and two indels were found in the more variable region. A request on Rfam (Kalvari et al., 2021) was not more successful. When submitted, the ‘complete’ 16S of H. covidum JL351 aligned with a 574 bp portion (out of 958 bp) of the 16S gene of the flatworm Stenostomum cf. simplex AW-2018, with an E-value of 3.4e−42 and an identity of 60.45%.

Figure 35 An alignment of the ‘complete’ 16S genes from all Bipaliinae displayed as a LOGO.

The alignment obtained from seven sequences representing six species shows the presence of a more conserved second part of the gene while the first part appears strongly variable.

Figure 36 Alignments of the ‘complete’ 16S genes from H. covidum JL090 and JL351.

The two specimens are from the French (JL09) and Italian (JL351) populations. The black star indicates the beginning of the most conserved part evidenced by multispecies alignment.

Phylogeny

The four phylogenetic trees displayed some variations in their topologies, impacted by the fact that the sampling of species was not identical for each of the phylogenies conducted. LSU was the most documented in this case. In the SSU tree (Fig. 37), H. covidum appeared as a sister-group to B. vagum, but with low support at the nodes (44% bootstrap in ML and 0.50 posterior probability in BI). This clade was, in contrast, strongly separated from the other clade containing B. adventitium, B. kewense, B. nobile, D. multilineatum, and N. venosum. In the LSU tree (Fig. 38), both H. covidum and B. adventitium were separated from the main clade of Bipaliinae, with a polytomy. The position of B. vagum was again the least supported of the tree (40% ML, 0.72 BI), and in this case, it was associated with the main clade. The mitochondrial protein tree (Fig. 39) showed the highest support. In this tree, H. covidum was associated with B. adventitium. Bipalium vagum was again distinct from the main clade, but here with 100% support. Finally, the three-gene tree (Fig. 40) also associated H. covidum with B. adventitium, and both with B. vagum, but with lower ML node supports (65% and 59%, respectively), while BI node supports were higher (1.00 and 0.96, respectively).

Figure 37 SSU phylogenetic tree of bipaliine geoplanids.

Maximum likelihood phylogenetic tree based on 14 partial SSU genes, using the GTR+I+G model of evolution, with the best tree out of 100 computed for 1,000 bootstrap replications. The tree with the best likelihood is shown (−2,551.353092). ML bootstrap support values on the left. The BI tree had and identical topology, posterior probabilities are indicated on the right as decimal values. Diversibipalium mayottensis n. sp. appears as the sister-group to all other bipaliines. The subfamilies within the Geoplanidae (Rhynchodeminae, Geoplaninae and Bipaliinae) are indicated. Diversibipalium mayottensis branch in bold to show its position as sister-group to all other Bipaliinae.

Figure 38 LSU phylogenetic tree of bipaliine geoplanids.

Maximum likelihood phylogenetic tree based on 20 partial LSU genes, using the GTR+I+G model of evolution, with the best tree out of 100 computed for 1,000 bootstrap replications. The tree with the best likelihood is shown (−4,759.571033). ML bootstrap support values on the left. The BI tree had identical topology; posterior probabilities are indicated on the right as decimal values. The subfamilies within the Geoplanidae (Rhynchodeminae, Geoplaninae and Bipaliinae) are indicated. Diversibipalium mayottensis branch in bold to show its position as sister-group to all other Bipaliinae.

Figure 39 Phylogenetic tree of concatenated mitochondrial proteins of bipaliine geoplanids.

Maximum likelihood phylogenetic tree based on concatenated protein sequences extracted from 19 mitogenomes using the mtART+I+G model, with the best tree out of 100 computed for 1,000 bootstrap replications. The tree with the best likelihood is shown (−4,759.571033). The subfamilies within the Geoplanidae (Rhynchodeminae, Geoplaninae and Bipaliinae) are indicated. Diversibipalium mayottensis branch in bold to show its position as sister-group to all other Bipaliinae.

Figure 40 Three-gene phylogenetic tree of bipaliine geoplanids, based on concatenated cox1, SSU and LSU genes.

Maximum likelihood phylogenetic tree based on 18 concatenated partial sequences of cox1, SSU and LSU, using the GTR+I+G model of evolution, with the best tree out of 100 computed for 1,000 bootstrap replications. The tree with the best likelihood is shown (−24,779.059136). ML bootstrap support values on the left. The BI tree had identical topology; posterior probabilities are indicated on the right as decimal values. The subfamilies within the Geoplanidae (Rhynchodeminae, Geoplaninae and Bipaliinae) are indicated. Diversibipalium mayottensis branch in bold to show its position as sister-group to all other Bipaliinae.

The most noticeable difference between the concatenated trees was the relative position of the Geoplaninae and Rhynchodeminae. In the mitochondrial protein-coding genes tree (Fig. 39), Bipaliinae were associated with Rhynchodeminae with a node support of 62%, while Geoplaninae were distinguished from both with a node support of 100%. The three-gene tree (Fig. 40) associated Bipaliinae and Geoplaninae with node supports of 68% ML and 1.00 BI, while Rhynchodeminae were distinguished from both with a node support of 68% ML and 1.00 BI. It must be noted that for the three-gene phylogeny, the cox1 partial gene only accounted for ca. 10% of the size of the trimmed concatenated sequences, since it had to include the partial genes of N. venosum and B. nobile, which were consequently shorter than the complete genes retrieved from full mitogenomes. Nonetheless, this difference in topology is intriguing, and would justify further investigations.

There was a constant and substantial result displayed by all phylogenies (Figs. 37–40), which is the position of D. mayottensis, always outside the main clade including all other Bipaliinae, with very high support. In contrast to B. vagum for example, whose position varied depending on the marker, D. mayottensis always appeared as a sister-group and relatively distant from a clade including all available representatives of Humbertium, Bipalium, Novibipalium and Diversibipalium. Diversibipalium mayottensis thus appeared to be the sister-group of all other bipaliines.

Alien DNA and prey

Positive results for alien DNA were obtained for B. adventitium, B. vagum and both specimens of H. covidum. All results are listed in Table 5, and are available as File S3 and discussed below.

Table 5 Alien DNA detected in the samples.

Sample and geographic origin	Contig size (in bp)	Coverage	Best blastn results (organism, accession number, E-value, identity)	
H. covidum JL090
Billère, France	5,953	87.595262	Arion hortensis, KU341315, 0.0, 99.19%	
2,209	20.638418	Discus rotundatus, FJ917212, 0.0, 98.28%	
2,150	2.882324	Arion hortensis, MG856341, 0.0, 99.77%	
1,624	72.587394	Discus rotundatus, FJ917212, 0.0, 95.57%	
H. covidum JL351
Casier, Italy	14,281	3.216540	Cochlicopa lubrica (cox1 only), MF544766, 0.0, 99.24%	
2,140	93.634550	Cochlicopa lubrica, AY014019, 0.0, 99.23%	
633	136.802920	Cochlicopa lubrica, GU331944, 0.0, 99.84%	
413	113.274390	Oxychilus alliarius, MN022707, 0.0, 97.64%	
312	125.061674	Various gastropods	
B. adventitium JL328
Montréal, Québec, Canada	9,730	52.980923	Eisenia foetida, AF212166, 0.0, 98.87%	
B. vagum JL307
Morne Vert, Martinique	5,994	7.978338	Subulina octona, MF444887, 0.0, 99.97%	
3,025	22.415306	Subulina striatella, MN022690, 0.0, 99.66%	

Gastropod DNA was found among both specimens of H. covidum. Depending on the megablast results, some of these sequences could be linked with known species. Results obtained on H. covidum JL090 (from France) suggest that this specimen has been feeding on the garden slug Arion hortensis (A. Férussac, 1819) (Arionidae). There were also traces of DNA possibly originating from Discus rotundatus (O. F. Müller, 1774) (Discidae), a very small species of land snails, although here the megablasts are to be interpreted with more caution regarding their percentage of identity. For H. covidum JL351 (from Italy), most of the sequences found suggest that it has been feeding on Cochlicopa lubrica (O. F. Müller, 1774) (Cochlicopidae), another species of small land snail. Among others, a large contig corresponding to a complete, circular mitogenome was found by additional data mining after retrieving its SSU. After trimming and extraction of its cox1 gene, a megablast query returned 99.24% identity with MF544766-Cochlicopa lubrica. For B. adventitium JL328, we found traces of a Lumbricidae. Finally, B. vagum JL307 (from Guadeloupe) had traces of DNA probably originating from Subulina octona (Bruguière, 1789) (Achatinidae) or Subulina striatella (Rang, 1831), two snail species widespread in the Caribbean.

Discussion

The new species Humbertium covidum

Molecular results: cox1 sequences of specimens from three localities

The partial cox1 sequences of the three specimens from the two localities in France were identical, suggesting that they belong to the same population. The two localities (Saint-Pée-sur-Nivelle and Billère) are distant by about 100 km. The cox1 sequences of all 6 specimens from Italy (a single locality) were identical. The partial cox1 sequences of the Italian specimens were different from the French specimens by 2.58%. We consider that these differences are intraspecific, and that the same species was involved in both localities (Fig. 1). A longer discussion is provided below, based on complete mitogenome sequences.

Morphology and systematics

The genus Humbertium was erected (Ogren & Sluys, 2001) to accommodate species (23 species stated but only 22 listed) with the single apomorphic condition OVD-1 in which the ovovitelline ducts turn dorsally before reaching the gonopore and having an antero-dorsal entrance to the female organ, the proflex condition. Currently, of the 22 species of Humbertium, excluding H. covidum, three species (H. ferrugineoideum (Sabussowa, 1925), H. sikori (von Graff, 1899), and H. palnisium (de Beauchamp, 1930)) are uncertain as the OVD-1 character is not clearly shown in figures or mentioned in the text (Ogren & Sluys, 2001). Only three species are well described: H. ceres (Moseley, 1875), H. ravenalae (von Graff, 1899), and H. woodworthi (von Graff, 1899), the descriptions of the remainder being too concise, or mostly confined to the external morphology and the anatomy of the copulatory organs.

The type-species of Humbertium is Perocephalus ravenalae von Graff, 1899. Externally, H. covidum mainly differs from this species with its brown-black to black dorsal ground colour and lacking dorsal stripes (H. ravenalae has a brownish dorsal ground colour with fine paired dark median stripe either side of a pale median stripe that passes onto the black headplate, and fine paired dark marginal stripes). The length of H. ravenalae is some three times that of H. covidum, and the body apertures are more posteriorly displaced. The internal anatomy of H. ravenalae was described by Mell (Mell, 1903; von Graff, 1899). Humbertium covidum shares the same pharyngeal musculature and pharynx type as H. ravenalae, the general musculature of the copulatory organs, and the near vertical placement of the female glandular canal, though in H. ravenalae the proximal female canal tilts anteriad, while in H. covidum it tilts slightly posteriad. A viscid gland and common genital canal of the type in H. covidum and H. ceres are absent in H. ravenalae.

In the two specimens of Humbertium covidum examined histologically, the ovovitelline ducts turn dorsally before the gonopore (holotype) and at the posterior lip of the gonopore (paratype), rise and enter the female glandular canal antero-dorsally. Despite the slight difference between the two specimens at the point at which the ovovitelline ducts begin to ascend, attributed here to relative differences in maturity, the antero-dorsal entrance of these ducts into the female canal are present in both specimens, and it is considered that they exhibit the OVD-1 condition that characterises species of the genus Humbertium.

Within the genus Humbertium, H. covidum is a small species about 20 mm long, readily differentiated externally from the only other described and considerably larger black species, H. ferrugineoideum (Sabussowa, 1925) from Madagascar, which attains a length of 75–80 mm, and is black both dorsally and ventrally (H. covidum is grey to greyish brown ventrally), with a white margin of the anterior headplate that is absent in H. covidum. Internally, the penis and female glandular canal of H. ferrugineoideum are both acutely angled ventrad some 20o from the vertical (the penis bulb is almost horizontal in H. covidum), the glandular canal is not thistle-shaped as in H. covidum, and there is no viscid gland (present in H. covidum).

Externally, plain brown-black to black H. covidum is distinguished from similar small “black” species. These include Diversibipalium piceum (von Graff, 1897 in von Graff, 1899) from central Sulawesi, that is 43 mm long (preserved) black with blueish stippling dorsally and ventrally with black creeping sole, and well developed lappets on the headplate (H. covidum has a reniform headplate without lappets, without blue stippling and with a pale brownish-grey to grey creeping sole). Similar small “black” species also include D. smithi (von Graff, 1899) from Darjeeling, northern India, 54 mm long (preserved) with velvety blueish black dorsum with a touch of dull brown, and yellowish-rusty brown colour ventrally with a deep cream-yellowish creeping sole demarcated with blueish-light green margins (von Graff, 1899, Whitehouse, 1914) (H. covidum lacks a blueish cast to the dorsal ground colour and has a pale brownish grey to grey creeping sole that is not demarcated as in D. smithi). Two other much larger species with dark brown to black ground colour are D. richtersi (von Graff, 1899) from Madagascar, 94 mm long (preserved) with a small head with weakly formed lappets, dark brown dorsally and ventrally, grading to a reddish colour under the headplate, and mouth displaced more posteriorly than in H. covidum, and D. kirckpatricki from Sri Lanka, 60 mm long (preserved), dark brown dorsally and ventrally, with a pale creeping sole, but with strongly recurved lappets as in D. falcatum from Sumatra, and mouth displaced more posteriorly than in H. covidum. There is also an alien black molluscivorous Diversibipalium species, some 110+ mm long (living) with rounded lappets and small brownish headplate, and possibly with a black median dorsal stripe, recorded in and around Durban in South Africa (Himansu Baijnath pers.com to LW 2016 and observations #35482045, #37914997 and #61592889 in iNaturalist); the species is considered too large to be H. covidum.

The specimens with external morphology nearest to H. covidum are the Diversibipalium sp. “Kumamoto” of Yamamoto (2000) from Japan that is 30 mm long, dark brown-black in colour with an indistinct dark mid-dorsal stripe (Yamamoto, 2000). However, the dorsal aspect of a living specimen of H. covidum is indistinguishable in photographs from that of an undescribed species of Diversibipalium from Xiamen, China (see Table 6 for iNaturalist data).

Table 6 Humbertium covidum: possible occurrence worldwide.

Locality, Country	Comments	Date	Reference	C/P	
Saint-Pée-sur-Nivelle, France		2013	Justine et al., 2018	C	
Billère, France		2019	This paper	C	
Casier, Province of Treviso, Italy		2019	This paper	C	
Kumamoto, Kyushu Island, Japan		2000	Kawakatsu, Sluys & Ogren, 2005	P	
Marina di Cerveteri, Province of Rome, Italy	Photographs from Citizen Science	2014	Mori et al., 2022	P	
Likander peninsula, Popov Island, Eugénie Archipelago, off Russia, Sea of Japan	Based on their Fig. 3	2017	Prozorova & Ternovenko, 2018	P	
Xiamen, Chinese coast facing Taiwan strait, P. R. China	iNaturalist observation	2018	Observation #19171303	P	
Xiamen, P. R. China	iNaturalist observation	2018	Observation #19171787	P	
Petrov Bay, Lazovsky Nature Reserve, Primorye Territory, Russia	Based on her Fig. 2A	2019	Prozorova, 2021		
Hachijō-jima Island, Philippines Sea, off Japan main islands	Based on their Fig. 2	unknown	Meyer-Rochow & Miinalainen, 2020	P	
Notes:

The table provides a list of possible occurrences based on similarity of external morphology. All these records need to be confirmed, especially by molecular methods.

C/P: C, confirmed with molecular data; P, possible, based on photographs.

Internally, with regard to the anatomy of the copulatory organs, in particular the morphology of the proximal female glandular canal, the unusual common genital canal, and presence of a viscid gland, Humbertium covidum stands closest to Humbertium ceres (Moseley, 1875), originally described from specimens collected in the Royal Botanic Gardens, Peradeniya near Kandy, Sri Lanka (Moseley, 1875), with the internal anatomy subsequently described by von Graff (1899).

Externally, a preserved specimen of H. ceres measures 79 mm in length, with mouth 52 mm (65.8% of body length), and gonopore 64 mm (81% of body length), both displaced more posteriorly than in H. covidum. In addition, the dorsum of H. ceres is divisible into five longitudinal stripes, the whole of the dorsal aspect of the planarian is irregularly speckled in black, and the headplate is ornamented in dark and light bands. The ventral surface is characterised by paired slight sub-marginal glandular ridges.

Internally, the copulatory organs of H. ceres share with H. covidum an unusual development of the genital pad creating a broad, narrow elongate common genital duct. At the anteriad end of the duct in H. ceres is what von Graff terms a uterus (von Graff, 1899). A similar structure, identified here from its secretions as a viscid gland, is present in the anteriad genital pad at the end of the common genital duct in H. covidum; it is highly likely the “uterus” of H. ceres is also a viscid gland. The thistle-shaped proximal end of the female glandular canal in H. covidum is similar to the shape of the seminal receptacle at the proximal end of the glandular canal in H. ceres. However, the seminal receptacle in H. ceres does not receive shell gland secretions, and the ovovitelline ducts open into the glandular canal below the receptacle. In H. covidum, the ovovitelline ducts enter the invaginated dorsal end of the proximal glandular canal that receives shell gland secretions. The major difference between these two species is the anteriorly prolapsed female glandular canal in H. ceres, characteristic of a group of three species in Humbertium: H. ceres, H. proserpina and H. woodworthi that all exhibit this feature (character FCA-2 (Ogren & Sluys, 1998)), absent in H. covidum in which the female glandular canal is almost vertical with a slight posteriad tilt.

The viscid gland in H. covidum is characterised by cyanophil secretions and appears analogous to the viscid glands described in species of Rhynchodemini and Caenoplanini (Winsor, 1998a; Winsor, 1998b). It differs from the musculoglandular organs described by Müller (Müller, 1902) in Bipalium graffi and B. bohmigi (Type III of Winsor (Winsor, 1998a)) that discharge erythrophil secretions into the common atrium. These musculoglandular organs are situated on the genital bulge and appear analogous to the adenochiren on the atrial flaps of species of Artioposthia in which they have been demonstrated to have a role in cocoon formation (Winsor, 1998a).

Occurrences in Europe and possible occurrences in Asia

As mentioned above, the species has been found in two widely separated gardens in the Department of Pyrénées-Atlantiques in the South-West of France, and one locality in the Province of Treviso in North-Eastern Italy. However, it is well known that bipaliine species are most numerous in South East Asia and Madagascar (von Graff, 1899); we found in the literature and citizen science databases a few records that might be the same species (Table 6). Most localities in Asia appear to be on islands or coastal areas, but the database is certainly extremely incomplete.

Humbertium covidum is probably a species originating from Asia and is an alien species in Europe. Whether it will become an invasive species needs to be monitored in the future.

The new species Diversibipalium mayottensis

Morphology

There are no other bipaliine planarians described with the blue-green iridescent dorsal ground colour observed in D. mayottensis. Similar iridescence, which is lost on fixation, has been observed in various species of Caenoplanini, and is possibly due to tightly packed transparent proteinaceous rhabdoids in the epithelium, acting as a diffraction grating (Winsor, 2003).

With regard to the club-shaped headplate and general body shape, D. mayottensis is similar to the general morphology of species of Humbertium. In particular, D. mayottensis shares the relative positions of the body apertures with the mouth present in the anterior second fifth of the body, and gonopore in the fourth body fifth, with two species: H. woodworthi (von Graff, 1899) with four dark dorsal stripes, from Madagascar, and H. subboreale (Sabussowa, 1925) a small dark brown species from China.

Molecular characteristics

In 2018 we wrote: “The COI barcode of this specimen is clearly different from all other known sequences. We can safely claim that this species has never been sequenced before” (Justine et al., 2018). Our current results on the complete mitogenome confirm that the species is distinct from all other species for which the mitogenome is known; in addition, D. mayottensis was sister-group to all other bipaliines in all our phylogenetic analyses. This is probably more significant than the superficial morphological resemblance with various Humbertium species mentioned above.

Possible origin of the species

Because of the proximity of Mayotte with Madagascar, it may be hypothesized that the origin of the species is Madagascar, not Asia as for most Bipaliinae.

Mitogenomes

Including B. kewense (Gastineau et al., 2019), there are now up to 6 species of Bipaliinae for which mitogenomes have been sequenced. For some of them, there were a few protein-coding genes for which it was not possible to find either start or stop codons. There are already several reports among Platyhelminthes of mitochondrial protein-coding genes for which no start codon could be found (Justine et al., 2020a; Ross et al., 2016; Sakai & Sakaizumi, 2012; Solà et al., 2015). In the case of H. covidum, the ND3 gene is supposed to have a premature stop, by addition at the 3′ extremity of two A after a T, immediately followed by the tRNA-Ala. No stop codon or premature stop could be found at all for the cob gene of B. adventitium JL328, for reasons that remain unknown. Excluding D. multilineatum because of its incompleteness, it is possible to say that most of these species have mitogenomes of a size similar to B. kewense (ca. 15,500 bp), with D. mayottensis being slightly longer (15,989 bp). The main exception is B. vagum, whose mitogenome is 17,149 bp long. Bipalium vagum also had the highest number of alternative start codons, with four protein-coding genes concerned (cox3, atp6, ND1, ND4L). This extra-length seems to be explained by large intergenic sections located between the 16S and cob genes, where the three conserved tRNA (tRNA-Leu, tRNA-Thr, and tRNA-Asn) are separated from each other by hundreds of base pairs. We could not circularise the mitogenome of D. multilineatum, even after several iterations of Consed’s ‘addSolexaReads’ function. This suggests that this lacking region consists of repeated sequences that short-read sequencing technologies fail to reveal. We underline the fact that this missing part is located at the very same position as the extra length in B. vagum’s mitogenome.

We would also like to indicate that recent investigations on parasitic flatworms such as Echinococcus granulosus Batsch, 1786, Clonorchis sinensis Loos, 1907 and Schistosoma haematobium (Bilharz, 1852) using long-read technologies have shown considerable extra-lengths within these mitogenomes, as much as 18.5 kb long (Kinkar et al., 2021; Kinkar et al., 2019; Kinkar et al., 2020). We tend to think that in the near future, long-read technologies might unveil similar features among Geoplanidae.

Alien DNA and diet

Our results on alien DNA suggest that H. covidum feeds on slugs and snails, with a very clear result concerning Cochlicopa lubrica in Italy; this is the only information currently available concerning the diet of this new species. The information was based on a small number of specimens and should be confirmed by additional experiments. Results on B. adventitium (from Canada) suggest that the specimen fed on a lumbricid earthworm, a result compatible with other information on the diet of the species (Ducey et al., 1999). For B. vagum JL307 from Guadeloupe, results suggest that the specimen fed on a species of Subulina, a small snail; the species is known to feed on snails (Ducey, McCormick & Davidson, 2007). Interestingly, similar studies on Amaga expatria, an alien geoplanid found in Martinique, another island in the Caribbean, also found that it fed on species of Subulina (Justine et al., 2020a); species of Subulina are widespread in the Caribbean (Delannoye et al., 2015).

A distinct genus for Diversibipalium mayottensis?

All phylogenies showed D. mayottensis as a sister-group to all other Bipaliinae, thus confirming its appurtenance to the subfamily, but making it impossible to assign it to any of the known genera of bipaliines. The subfamily currently includes four genera, namely Bipalium, Humbertium, Novibipalium Kawakatsu et al., 1998 and the collective genus Diversibipalium. External morphology superficially suggests that the species is close to Humbertium, but the reproductive anatomy is unknown. Its position as sister-group to all other bipaliines suggests that a new genus should be described to accommodate D. mayottensis. We refrain from doing so here in the absence of anatomical information.

Conclusion

In this paper, we formally described two species of bipaliine geoplanids, previously only known as unnamed species included in the collective genus Diversibipalium. For the first species, Humbertium covidum n. sp., we subsequently obtained fresh specimens collected in Italy and could fully describe the anatomy, based on histological methods. This was not possible for the second species, found only on Mayotte, which is described here as Diversibipalium mayottensis n. sp. on the basis of external morphology. We newly characterised the complete mitochondrial genome of five species of bipaliine geoplanids, including the two new species and B. adventitium, B. vagum and D. multilineatum. Based on phylogenetic analyses of the SSU, LSU, mitochondrial proteins and concatenated cox1-SSU-LSU, we built phylogenies of bipaliines for which these sequences are available (6 species). In all phylogenies, D. mayottensis was the sister-group of all other bipaliines, suggesting that it represents a distinct genus, which needs formal description; this will await availability of additional specimens. Furthermore, we demonstrated that next generation sequencing methods provide an excellent tool for delineating and describing species of geoplanids, since they allow access to both traditionally used sequences (SSU, LSU and cox1) and complete mitochondrial genomes which provide considerable additional information.

Supplemental Information

Supplemental Information 1 Specimens of Humbertium covidum n. sp. and Diversibipalium mayottensis n. sp. examined for the present study.

Click here for additional data file.

Supplemental Information 2 Humbertium covidum alive.

Specimen crawling on flat surface. The scale shown in the last seconds is graduated in millimetres. Video by Enrico Ruzzier, modified by Jean-Lou Justine.

Click here for additional data file.

Supplemental Information 3 Alien DNA sequences from prey.

"Alien" DNA sequences detected in various samples. They correspond to the prey inside the digestive tract.

Click here for additional data file.

Supplemental Information 4 Mitogenomes deposited in GenBank but not available yet.

6 complete mitogenomes sequences deposited as MZ561467–MZ561472.

Click here for additional data file.

We thank the colleagues and people who provided specimens, especially Mathieu Coulis, Mathieu Théry, Geneviève Rolland-Martinez and Dino Carraro. We emphasize that lockdowns and social distancing helped us to concentrate on completion of this paper, but we will not forget that the pandemic has affected and still affects the world terribly–the Latin epithet covidum for our new species should thus be considered a homage to the victims of the COVID-19 pandemic.

Abbreviations used in figures of histology

af atrial flap

ca common atrium

cc copulatory canal

cgc common genital canal

ch chondrocytes

clm cutaneous longitudinal muscles

cm cutaneous musculature

cs ciliated creeping sole

dfg distal female glandular canal (= vagina)

dip dorsal insertion of pharynx

dtm dorsal transverse muscles

ed ejaculatory duct

eg erythrophil glands

pfg proximal female glandular canal

g gonopore

gm glandular mesenchyme

gp genital pad

i intestine

ma male atrium

m mouth

nc nerve cord

ovd ovovitelline duct

pb penis bulb-penis

pg penial glands

ph pharynx

php pharyngeal pouch

pp penis papilla

sd spermiducal vesicle

sg shell glands

sr seminal receptacle

sv seminal vesicle

te testis

tm transverse parenchymal muscle

vd vas deferens

vg viscid gland

vi vitellaria

vip ventral insertion of pharynx

vp ventral muscle plate

Additional Information and Declarations

Competing Interests

Author Contributions

DNA Deposition

Data Availability

New Species Registration

Jean-Lou Justine is an Academic Editor of PeerJ.

Jean-Lou Justine conceived and designed the experiments, performed the experiments, analyzed the data, prepared figures and/or tables, authored or reviewed drafts of the paper, obtained funding, coordinated collective effort, and approved the final draft.

Romain Gastineau conceived and designed the experiments, performed the experiments, analyzed the data, prepared figures and/or tables, authored or reviewed drafts of the paper, performed bioinformatics on mitogenomes and phylogenetic analyses, and approved the final draft.

Pierre Gros performed the experiments, prepared figures and/or tables, made photographs of living animals, and approved the final draft.

Delphine Gey performed the experiments, authored or reviewed drafts of the paper, performed Sanger sequencing, and approved the final draft.

Enrico Ruzzier performed the experiments, prepared figures and/or tables, authored or reviewed drafts of the paper, collected worms, made a movie, and approved the final draft.

Laurent Charles performed the experiments, prepared figures and/or tables, authored or reviewed drafts of the paper, collected worms, made photographs, and approved the final draft.

Leigh Winsor conceived and designed the experiments, performed the experiments, analyzed the data, prepared figures and/or tables, authored or reviewed drafts of the paper, performed histology, made drawings and photographs, and approved the final draft.

The following information was supplied regarding the deposition of DNA sequences:

The sequences are available at GenBank:

MG655588, MZ520996, MZ622153, MZ622148–MZ622152, MZ647546–MZ647548, MZ520988, MZ520995, MG655598, MG655596, MG655597, MG655599, MZ561467–MZ561472.

The following information was supplied regarding data availability:

The six following sequences (complete mitogenomes) are available in the Supplemental File and at GenBank: MZ561467–MZ561472.

The partial sequences from “alien” DNA (DNA from preys) are available in the Supplemental File.

The following information was supplied regarding the registration of a newly described species:

Publication LSID: urn:lsid:zoobank.org:pub:27A4D685-9042-40C2-A40A-89FF8BCC489B.

Humbertium covidum n. sp.

urn:lsid:zoobank.org:act:3847E9FE-463B-4FDB-A164-88765A52D65A.

Diversibipalium mayottensis n. sp.

urn:lsid:zoobank.org:act:B59FEE8E-70FD-4DEC-B839-554C351701F8.

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
