# Peer review of "Hammerhead flatworms (Platyhelminthes, Geoplanidae, Bipaliinae): mitochondrial genomes and description of two new species from France, Italy, and Mayotte"

_PeerJ, doi:10.7717/peerj.12725_

## Round 0.1 · original submission · Major Revisions

Dear authors,

Now I have two reviewer’s comments and attending to their advice I want to congratulate you for a great work done, which for sure qualifies to be published in PeerJ. Nonetheless, both reviewers find a series of issues that need to be tackled, although one classifies them as minor I agree that given some of the issues will require doing some new analyses (nothing that can not be done easily) I better qualify them as major revisions.

Please read carefully both reviews that I think are well detailed and mostly clear on what they ask. In general, they coincide in their appreciations, so I think it will be easy for you to give answer to all their queries.

Issues I find especially important:

From reviewer 1 comment on Material and methods line 153. Please follow the advice and redo that tree.

Also very important is reviewer 2's second comment on material and methods (line 160).

Both reviewers also coincide it will be important to try other inference methods and also do an analysis of a concatenated data set (reviewer 1, lines 544 and 548; reviewer 2, comments 8 and 10).

Both reviewers also suggest using other programs to search for tRNAs. And also, it is necessary to explain how you did decide the evolutionary model to be applied as they ask (either if it was using a program or ad hoc).

I also completely agree with the comments of both reviewers referring to your introduction (line 95 or line 90 attending to one or the other reviewer), please modify.

Also reviewer 2 points to an important issue on the conclusions in comment 1. I totally agree with that comment, and also like the extra conclusions the reviewer proposes to add (not related to the previous one but that can be clearly drawn from your analyses).

Although I have highlighted these issues as important, and especially because on most of them both reviewers coincide, please attend carefully to all the rest of the issues raised by both reviewers. I’m certain the quality of the manuscript, already of great value, will increase.

Reviewer 1 ·

Basic reporting

Fine

Experimental design

Fine

Validity of the findings

Fine

Additional comments

The article "Hammerhead flatworms (Platyhelminthes, Geoplanidae, Bipaliinae): mitochondrial genomes and description of two new species from France, Italy and Mayotte" is an interesting contribution to the taxonomy of terrestrial planarians of the bipaliinae group. Two new species are described and complete (or nearly complete) mitogenomes are provided for several species in the group. The article is robust and well thought out, however, some aspects of the methods could be improved to achieve more accurate results. The figures are of good quality, although some are not necessary for the main text. The quality of the aesthetics of phylogenetic trees can be improved. I have to say that the integration of the molecular part in the description of the species seems very successful to me and for this, I congratulate the authors.
In summary, I consider that the article can be accepted for publication in PeerJ with some minor changes. Below I comment in more detail on some aspects that I think should be improved.

Introduction
-Line 95: “Complete mitogenomes provide far more data for molecular identification and phylogeny than, for example, the usual molecular markers such as the short and long subunits of the nuclear ribosomal RNA genes (SSU and LSU, respectively).” I disagree with this affirmation. It depends on the level of the phylogeny you want to solve. Sometimes mitochondrial genomes don’t give information enough to point out relationships at higher taxonomic levels (not species).

Material and methods
-Line 153: Why neighbor-joining? It is known that the accuracy of phylogenies using this method is questionable. Although probably the result would be the same, I encourage the authors to use another inference method for this analysis (as they have done in other sections). To maintain the consistency of the analyses across the article it makes sense to use the same strategy for all the phylogenetic inferences (unless there is a particular reason for using a different method here, that if is the case, needs to be justified).
-Line 159: Why the authors chose this k-mer? Did they use any software to estimate the best k-mer?
-Line160: How did the authors assess which contigs corresponded to mitogenome? Did they apply any filter to detect possible contamination? They talk about contamination later in the results and discussion, but nothing is noticed in the material and methods section. It should be explained here.
-Line 176: How was the model chosen? Based on what? Please, add this information to the text.

Results
-Do the authors of the species correspond to the same authors in the article? That is not always the case, and if so, it must be stated in the description of the species for further citations.
-As I already said before, I like the molecular section added to the description of the new species, but maybe it lacks some explanation of how to identify this species using molecular methods (like inferring a phylogeny or calculating genetic distances). This is only a suggestion, but I really think it would improve the quality of the descriptions (in general) if molecular information is added to the diagnosis of the new species.
-Line 480: It is confusing, there is a sexual specimen? Why is it not described here? If the case is that it is not sexually mature, it might be stated here, and a more accurate description of the external morphology is needed to classify the species within the Bipaliinae.
-Line 489: Was D. multilineatum mitogenome reads of lower quality? Why this is not complete? Was the circularization not possible? Maybe using Novoplasty or another software it is possible to get a better assembly.
-Line 495: How were the tRNA searched? What software did the authors use? It is not correctly explained in the M&M section. If so, it means that the authors only used MITOS for the annotation. There are specific programmes used for the tRNA search (such as tRNAscan, for instance). Applying these, maybe the authors can assemble it completely.
-Line 499: Information about the genes (length in comparison with other species, direction, start and stop codons, etc) is missing, as well as some remarks about the control region and intergenic sequences. Then in the discussion, some of these characteristics are mentioned, but this information needs to be described in the results section before.
-Line 519: When trying to solve the 16S problem, why not comparing to other terrestrial planarians? Schmidtea mediterranea is very divergent in terms of the mitochondrial genome, and maybe not the best to compare to (even being verified by RNAseq). An alignment with other species of Tricladida is another option to find out the best annotation for the 16S. Again, there are other programmes that can be used for annotation. Moreover, the 16S gene is one of the most frequently used for metabarcoding analyses, so there are complete databases that can be used for blast and find out the best annotation for the species studied here. Also, the secondary structure of the ribosomal gene can be used. There are plenty of possibilities.
-Line 544: Have the authors thought about using other inference methods to try to solve this polytomy?
-Line 548: Maybe a concatenated tree (including the two ribosomal genes and the mitogenomes) could solve all the nodes. It is worth trying.
-Line 575: Only using megablast is not accurate enough to have a complete picture of the diet of the species. Applying a pipeline based on phylogeny would be more accurate and give more information (see Cuevas-Caballé et al. 2019 for an example). Also, these contaminations make less trustable the validity of the flatworms mitogenomes. How can the authors be sure that the mitogenomes correspond to flatworm 100% and are not chimeras? As I asked before, were the raw data filtered and searched for contamination? This is not explained in the M&M section. If not, I wonder if that would be one of the reasons for the poor annotation of the 16S or the tRNAs.


Discussion
-Line 696: This part corresponds to the description of the results and not to the discussion.

Figures and tables
-Figures with pictures of the animals are impressive. Very good quality.
- The figures with images of the histological sections are not all necessary in the main text. Since a reconstruction of the copulatory apparatus is already presented, only a few photographs representing the most important anatomical and key characters can be left in the main text. The others can go to supplementary material.
-Table 1 can go to supplementary material. It lacks the information of the authors of the new species.
-Table 3 could have more information, for instance, which are the alternative start and stop codons (in addition to mentioning in which genes they are found).
-I suggest that Table 6 could be converted to a distribution map, which would give more visual information.

Reviewer 2 ·

Basic reporting

The authors of the paper “Hammerhead flatworms (Platyhelminthes, Geoplanidae, Bipaliinae): mitochondrial genomes and description of two new species from France, Italy and Mayotte” describe the first Bipaliinae species in about 15 years, and among them the first Humbertium species in 35 years. Furthermore, the authors also provide 6 new mitochondrial genomes of 5 different species, giving much valuable new information to the study of this subfamily of land planarians.

Overall, the paper is well-written and well-organized. Literature references are sufficient.

Experimental design

Research is relevant, clearly filling knowledge gaps in the study of land planarians.

Some aspects of the material & methods can be better explained. On the other hand, some extra analyses could be carried out to improve this paper. Please see "Additional comments" for suggestions.

Validity of the findings

Please see "Additional comments" section.

Additional comments

The paper is interesting and I look forward to its publication. However, I would like to suggest some improvements and make some comments that might be helpful:

1- Line 50:
Here it is said that “Complete mitochondrial genomes provide a powerful tool for delineating…species of Bipaliinae when the reproductive structure cannot be studied.”. However, it seems not to be a conclusion that can be easily drawn from the presented results. The species studied in this paper do not represent a problematic case of externally similar species, but on the contrary they present very distinctive external appearances (colourations and patterns) which allow their rapid identification with the naked eye. On the other hand, in the present paper no molecular species delimitation analyses have been carried out.

I suggest a couple of possible conclusions:
1) The results seem to reinforce the idea of a very successful group of land planarians in colonizing areas beyond their native geographical range, not only in extension (e.g., Bipalium kewense being a Cosmopolitan species) but also in the diversity of species. Most known genera of Bipaliinae are found in Europe (i.e., Bipalium, Diversibipalium and Humbertium).
2) The phylogenetic position of Diversibipalium mayottensis suggesting a new Bipaliinae genera could be also included in the conclusions, pointing to more genera of this group to be described.

Obviously, different conclusions can be drawn by the authors.

2- Line 66-68:
It seems that the order of the two last sentences should be changed.

3- Line 90:
Although it is certainly true that complete mitogenomes provide lots of valuable data, I do not agree with the sentence: “Complete mitogenomes provide far more data than…nuclear ribosomal RNA genes”. Both types of markers provide valuable data, and both will likely fail in providing accurate phylogenies if analysed alone. For example, in the phylogenetic tree obtained using the mitochondrial protein-coding genes (Figure 39 of the present paper), Dugesia ryukyuensis is shown to be sister species to Dugesia japonica plus Girardia sp. with full support. In the case the authors decide to keep the original sentence, please add references.

4- Line 99:
“…for the first time the mitochondrial genomes of three species…”. However, the mitochondrial genomes of five species are presented for the first time in this paper. Is this sentence correct?

5- Line 141:
The obtaining of Cox1 and 28S is here explained, but nothing is said about the obtaining of 18S (nor in any Material & Methods section). On the other hand, somewhere else in the manuscript it is said that some 18S and 28S were obtained using NGS (e.g., line 205). Please include the information of the obtaining of the genes in this section comprehensively.

6- Line 141-146:
To help the reader, first time mentioning 18S and 28S please add (SSU) and (LSU) respectively as both ways of naming these genes are used along the manuscript.

7- Line 148:
Please specify in the text the amount of tissue that was sent. It was a single specimen? A piece? Multiple specimens?

8- Line 157:
It would be interesting to include a phylogenetic tree obtained from a concatenated dataset using 18S, 28S and Cox1 (the most widely used mitochondrial gene in phylogenies). The combination of these genes may provide a more robust phylogeny in comparison with the phylogenetic trees already presented in the paper. The concatenated dataset may include the species described in the present paper plus others from GenBank (e.g. Novibipalium venosum and Bipalium nobile). Both maximum likelihood (e.g. RaxML) and Bayesian Inference (e.g. MrBayes) should be used.

9- Line 159:
Please include a table listing all the specimens used in the different phylogenies/datasets on the first column, and the genes used in the first row, filling the boxes with the corresponding accession numbers.

10- Line 165:
Please consider carrying out Bayesian Inference analyses (e.g., using MrBayes) to provide a much complete phylogenetic approach. Only one topology per dataset is necessary to be shown in the paper but including both supports (Posterior Probability/Boostrap) on the corresponding nodes.

11- Line 167:
GTR-I-G was chosen as evolution model. Please explain in the text whether it was chosen after running jModelTest or a similar program.

12- Line 194:
If possible, it would be useful to include an “Habitat and distribution” section for this species explaining when and where in the garden it was found. For instance: was it found crawling on the ground? In plant pots? During the day?

13- Line 196-198:
It is said that microslides of two specimens of the new Humbertium species are available (MNHN JL351B and MNHN JL351A). However, only one reconstruction is presented in the article (MNHN JL351B; Figure 17). Was a sketch of the other specimen (MNHN JL451A) made? If this is the case, it would be interesting to include it in the paper.

It would be helpful to show the intraspecific variability of the new species in figures. For instance, as it is said in the text the oviduct turning dorsal seems to be different in the two specimens. Photos showing these differences could be also used as an alternative to a full copulatory apparatus reconstruction drawing.

14- Line 196:
Was any of the specimens used for inner anatomy inspection also sequenced (i.e., MNHN JL351B and MNHN JL351A)? For instance, by cutting off a small tail or pre-pharyngeal piece for DNA before sectioning. If this is the case, please explain in the manuscript. If this was not done, I would like to suggest doing so in future studies. In this way, we can be 100% certain of the assignation of the inner anatomy to the molecular data.

15- Line 198:
I would like to ask whether “retained whole” means that the specimen has not been either sectioned or sequenced. This is confusing to me because specimen MNH JL351C is listed in this line as “retained whole” but photos of MNH JL351C sections are provided in Figures 19-21 (see figure descriptions). On the other hand, the number of microslides obtained for MNH JL351C is not provided. Furthermore, MNH JL351A is reported in line 197 as sectioned, but no photo or reconstruction is provided.

16- Line 198:
I would also like to ask to the authors what is the difference between “…MNH JL351C-4 retained whole…” and the following “…9 other specimens retained whole.”? Why do the latter not have voucher numbers? If necessary, please explain in the manuscript.

17- Line 200:
MNHN JL090 specimens do not have a specific voucher number (e.g., A to F) but sequences were obtained from them instead. Please see next comment.

18- Line 204-207:
According to line 200, MNHN JL090 is constituted by 6 specimens. It is not explained whether the Cox1, partial 28S, partial 18S and the complete mitogenome were obtained from the same or from different individuals. It would be necessary to assign each sequence to its original specimen.

I suggest including a table listing all the individuals assigned to the new Humbertium species, also indicating which genes and/or mitogenome (with Accession Number) per specimen were obtained, and whether inner anatomy has been inspected (i.e., microslides available). Other information like the locality could also be added.

19- Line 208:
Similarly to the comment above, MNHN JL351 specimen is not specified (A, B, C), and different genes are listed. Please clarify.

20- Line 212:
I would like to suggest finding a different species name, keeping the homage to the victims of the pandemic.

21- Line 217:
Here specimens that are not listed in the previous section are mentioned: MNHN JL351H, J, K, L, M. This could be clarified in the table I suggest preparing in comment 18.

22- Line 231-232:
The anatomy of the pharynx is included in the diagnosis of the new Humbertium species. Has any difference been observed between the pharynx of this species and that of other Bipaliinae? In the discussion, the pharynx is only compared with that of Humbertium ravenalae and it appears to share the same structure. In case the pharynx has not been found to be significant to identify the new Humbertium species, I suggest removing this sentence from the diagnosis section.

23- Line 236-237:
It would be convenient to briefly explain here what the differences of the “two parts” of the female genital canal are.

24- Line 228 & 252
When looking at the live Humbertium photos (Figures 2-10) the dorsal colouration seems rather dark brown or perhaps deeply black-brown instead of “pure” black. The ventral side seems paler or grayish brown in colouration instead of grey. Perhaps this might be explained by colour differences between the preserved and the live specimens. In the case the authors agree, please discuss in the text.

I would also like to say that the Humbertium photos are magnificent. It is wonderful that in some of them the creeping sole in action can be clearly seen (Figures 3 and 8, hardly in figure 7).

25- Line 257
If possible, it would be interesting to explain whether differences considering rhabdoids and secretions were observed between the head and the rest of the body.

26- Line 329-331
It seems that the outer penial epithelium underlying circular muscles depicted in figure 17 are not mentioned here. These circular muscles seem to be continuous with that “…inner strong sheath of circular muscles…” mentioned in the previous paragraph.

27- Line 354
Please add the corresponding paratype voucher number in parentheses.

28- Line 354
If possible, it would be interesting to show the “…ovovitelline ducts turn dorsally…(…in the Paratype they turn dorsally at the posterior lip of the gonopore),…” either in a reconstruction of its copulatory apparatus or in different sections photos, as I already suggested in a previous comment.

29- Line 391:
Perhaps a photo of the nematode might be helpful in the future to a specialist to narrow down its assignment. Could it be provided in the paper?

30- Line 420:
As I already suggested for the new Humbertium species, it would be good to include an “Habitat and distribution” section also for Diversibipalium mayottensis.

31- Lines 420-435:
The specimen from which the mitogenome was obtained seems not to be listed. According to figure 30 legend, the specimen was JL381. Perhaps it is a “typo” and it should be JL281. Nonetheless, there are 3 specimens from this locality (JL281A, B, C). Please indicate in the text and in a table (as suggested below) which is the corresponding one.

32- Lines 420-435:
According to the phylogenetic trees, 18S and 28S were obtained for D. mayottensis. However, only Cox1 sequences are mentioned in this section. As suggested for the other new species, please include a table listing all the specimens assigned to this new species, also indicating which genes and/or mitogenome (with Accession Number) per specimen have been obtained.

33- Line 460:
The mitochondrial gene ATP8 is not mentioned in the manuscript. Although it is true that it was widely said that this gene was not present in the Platyhelminthes mitochondrial genomes, Ross et al., 2016 (paper already cited here) and Egger et al., 2017* pointed otherwise. Ross and collaborators found an ATP8 candidate in different species of freshwater Triclads, including Dugesiidae (closely related to Geoplanidae). I suggest to carry out an active search of this gene in the mitochondrial genomes included here by aligning with other triclads’ ATP8 candidates and looking for further ORFs. Either successful or not, please explain the results in the manuscript.

*Bernhard Egger, Lutz Bachmann, Bastian Fromm, 2017. Atp8 is in the ground pattern of flatworm mitochondrial genomes. BMC Genomics, 18:414.

34- Line 469:
Here it is said that “…it was impossible to find tRNA-Thr for both specimens…”. I suggest trying programs other than MITOS, like ARWEN, tRNAscan-SE and/or DOGMA to find the missing tRNA and double-check the others.

35- Line 514-516:
In this sentence it is said that “This clade (i.e., H. covidum+B. vagum) was, in contrast strongly discriminated…”. However, supports for H. covidum+B.vagum and its sister group are low (44 and 68 respectively). Please explain whether “strongly discriminated” means something different.

36- Line 522-527:
Second and third sentences of this paragraph seem repetitive with the first one. Please combine them.

37- Line 563:
Please name the “…three species are uncertain as the OVD-1 character is not clearly shown in figures or text” in parentheses.

38- Line 563:
Please list the four “fully described” species in parentheses.

39- Lines 565-572:
The inner anatomy of the new Humbertium species is compared with the type species of the genus, but nothing is said about the external appearance. Please compare in the manuscript.

40- Lines 573-579:
It seems that this paragraph is misplaced and should be placed before the previous one.

41- Lines 580-581:
Here it is written: “…readily differentiated externally from the only other described plain black species, H. ferrugineoideum (Sabussowa, 1925) from Madagascar…”. Please explain in the text:

1) Which characteristics are readily differentiated.
2) Whether “Plain black species” refers to both dorsal and ventral sides.

Note also that Humbertium ferrugineoideum may not be considered plain black as it presents a white line along the head margin. On the other hand, please consider that H. ferrugineoideum is described as black both dorsal and ventrally while the new Humbertium species seems to be “black” (or dark brown or deeply black-brown as in comment 24) only on its dorsal side.

42- Line 582-583:
Please detail in the manuscript the “...substantial differences in the anatomy of the copulatory organs…”.

43- Lines 584-589:
I suggest a couple of other black/dark Diversibipalium species to be compared with:
- Diversibipalium kirckpatricki (Von Graff, 1899) – Deeply black-brown dorsal and ventral colour, with a lighter brown creeping sole. This might be the most similar known species to the new Humbertium species.
- Diversibipalium piceum (Von Graff, 1899) – Uniform black dorsal colour (but slightly blue).
Perhaps other dark brown species like Diversibipalium richtersi (Von Graff, 1899) could also be considered for comparison.

44- Line 627-628:
“…it is well known that bipaliines originate from Asia;…”: I would like to ask whether the authors are writing about the new Humbertium species or about the Bipaliinae subfamily in general. In the latter case, references are needed here. Species of Bipaliinae are also present in Madagascar and I think it is yet unknown which is the geographical origin of the subfamily.

45- Line 711-712:
Same than comment 1.

46- Lines 716-719:
This part seems not to fit well in the “Acknowledgements” section.

47- Figures 17-23:
Please indicate in the figure legends where “anterior” is (to the left?).

48- Figure 30:
Figure legend: Is it JL281 instead of JL381?

49- Figure 38:
DQ665957 and HM346595 are written twice.

---

## Round 0.2 · Minor Revisions

The paper is accepted provided that you have into account the minor revisions raised by one of the reviewers. I'll personally check and immediately accept the paper provided the changes have been done.

Best

Marta

Reviewer 1 ·

Basic reporting

The article is well written, understandable, and has good scientific English. The structure is correct and conforms to PeerJ standards.

Experimental design

No comment

Validity of the findings

No comment

Additional comments

The authors have made necessary changes, as indicated in the review, considerably improving the quality of the article. I agree with most of the answers to my comments and those of reviewer 2. I consider that now the article is ready to be published.

Reviewer 2 ·

Basic reporting

Ok

Experimental design

Ok

Validity of the findings

Ok

Additional comments

I appreciate the great effort by the authors in order to improve the article. I would like to congratulate them about that.

Just few minor points:

- Results of Bayesian Inference (Posterior Probability (PP)) are not percentages (%). Please correct along the text (e.g. Phylogeny section). I suggest showing the results in the manuscript in the following way: 68 BS/1.0 PP. Being BS = Bootstrap and PP = Posterior Probability.

- In the Taxonomy section of the two newly described species the authorship of each rank is written twice (e.g. Order Tricladida Lang, 1884 (Lang, 1884)). Is this correct?

- Line 43. It seems that there is an extra “-“ after “The type-…”.

- Line 83. Please name the “one species”.

- In Line 218 it is said that “The single genes and concatenated sequences were aligned using MAFFT7”. This is confusing to me. Were the genes first concatenated and then aligned or the other way around? Please note that the latter would be the correct procedure.

- Line 668. “The four phylogenetic trees displayed some variations in their topologies, impacted by the fact that the sampling of species was not identical for each of the phylogenies conducted.” Please also take into account that different genes may give different topologies even if the specimens are exactly the same for all of them.

---

## Round 0.3 · accepted · Accept

Dear Jean-Lou,

The manuscript is now accepted. I understand the addition of the new authors. I must nonetheless say that on the point referring to the alignment and concatenation of genes as it is explained in the ms seems done in a wrong way. As explained in the manuscript, it appears that you align individual genes on one side, and on the other, you concatenate the genes (before aligning) and then do an alignment for this concatenated dataset, instead of concatenating the aligned genes. Concatenating before aligning is a wrong procedure (since the alignment program does not know the limits of the genes and could align the final part of one gene with the initial of the following for some sequences) as the reviewer tried to explain, I imagine the issue is only a question of not having explained correctly the procedure in the material and methods.

Best regards,

Marta